# Dislocation interactions during plastic relaxation of epitaxial colloidal crystals

Ilya Svetlizky [1,4] ✉, Seongsoo Kim[1,4], David A. Weitz [1,2,3] & Frans Spaepen[1] ✉

The severe difficulty to resolve simultaneously both the macroscopic deformation process and the dislocation dynamics on the atomic scale limits our understanding of crystal plasticity. Here we use colloidal crystals, imaged on the single particle level by high-speed three-dimensional (3D) confocal microscopy, and resolve in real-time both the relaxation of the epitaxial misfit strain and the accompanying evolution of dislocations. We show how dislocation interactions give rise to the formation of complex dislocation networks in 3D and to unexpectedly sharp plastic relaxation. The sharp relaxation is facilitated by attractive interactions that promote the formation of new dislocations that are more efficient in mediating strain. Dislocation networks form fragmented structures, as dislocation growth is blocked by either attractive interactions, which result in the formation of sessile dislocation junctions, or by repulsion from perpendicular segments. The strength of these blocking mechanisms decreases with the thickness of the crystal film. These results reveal the critical role of dislocation interactions in plastic deformation of thin films and can be readily generalized from the colloidal to the atomic scale.

Atoms in crystalline materials are arranged in a perfect periodic order. Plastic deformation, which requires breaking this order, is mediated by nucleation and motion of topological line defects in the crystalline structure called dislocations[1]. Due to the complexity of dislocation interactions, the collective behavior of these defects remains one of the principal challenges of materials and statistical physics[2]. The wide range of time and length scales makes numerical modeling of dislocation dynamics computationally demanding[3,4]. Of particular value, therefore, are fundamental experiments on simple systems in which all the elements (stress, strain and dislocation configurations) can be closely controlled and observed.

Here we focus on the mechanisms by which dislocations are formed in thin films. It has been observed that nucleation and growth of dislocations relaxes the elastic strain induced by the lattice-mismatched substrate, if the crystals are grown above a critical thickness[5–7]. The early stages of the relaxation process are well understood[5,8,9], as dislocations are well separated and their interactions can be ignored. During the later stages of relaxation, dislocation interactions play a crucial role; however, determining the interaction mechanism presents a significant challenge, with proposed mechanisms giving contradictory predictions[9–14]. Our ability to identify the appropriate mechanism is limited by the difficulty to image in real-time both the relaxation of strain and the full 3D structure of the dislocation networks in atomic crystals.

Greater insight into thin film dislocation dynamics is also technologically important[15,16]: On the one hand, dislocations have detrimental effects such as decreasing electrical conductivity[17], photoconductivity[17], ferroelectricity[18], thermoelectricity[19], and photonic band-gap[20], while on the other hand, they have been used to advantage to control the functionality of thin films, enabling enhancement of superconductivity[21,22], switching of electrical resistance[23], and fabrication of ordered nanostructures[24] and nanoscale ferromagnetic elements[25].

[1]School of Engineering and Applied Sciences (SEAS), Harvard University, Cambridge, MA, USA. [2]Department of Physics, Harvard University, Cambridge, MA, USA. [3]Wyss Institute for Biologically Inspired Engineering, Harvard University, Cambridge, MA, USA. [4]These authors contributed equally: Ilya Svetlizky, Seongsoo Kim. ✉e-mail: isvetlizky@technion.ac.il; spaepen@seas.harvard.edu

Here we use thin colloidal crystals to determine the interplay between dislocation interactions and plastic relaxation. The advantage of colloidal crystals lies in the size of the particles, ~1μm. On the one hand, the particles are large enough so that they can be visualized by optical microscopy and on the other hand, the particles are small enough so that their thermal motion gives rise to a non-zero elastic crystal stiffness[26]. This advantage has been exploited extensively to study dislocations in two-dimensional lattices[27–31]. In three-dimensional (3D) crystals, however, the topology of dislocations fundamentally modifies the nature of their interactions. Here, we use high speed confocal microscopy to image 3D colloidal crystals in real-time and on the single-particle level. Crystals are strained by growing them on mismatched templates[32–35], in direct analogy with epitaxial atomic thin films. We resolve the relaxation process by direct measurements of the elastic strains and reconstruction of the 3D dislocation networks. The combination of the two reveals how pairwise dislocation interactions can, on the one hand, enhance the relaxation process and give rise to unexpectedly sharp plastic relaxation, and on the other hand, hinder dislocation motion and lead to complex dislocation networks. These results are key to our fundamental understanding of dislocation interactions in thin films and can readily be mapped to the atomic scale, given the topological nature of dislocations.

## Results

We disperse silica particles with a diameter $2R = 1.55\,\mu m$ in an index-matched fluid with Fluorescein-NaOH dye and control the Debye screening length of the particle solution by adding NaCl. Thin colloidal crystals are grown over an area of $1\,cm^2$ to a height of $h = 55\,\mu m$ at $5\,\mu m$ per hour, by sedimentation of the particles on either flat or templated substrates. We visualize in three dimensions five well separated regions of volume $200 \times 200 \times 60\,\mu m^3$ every 7–15 min, using a spinning-disk confocal microscope. Particle positions are then obtained by processing the confocal images. Further details are discussed in the Methods section.

As a reference lattice for crystal growth on a template, we first grow one on a flat substrate. The two simplest crystalline structures

that can be formed by stacking closed packed hexagonal layers are face-centered cubic (fcc), with *ABCABC* stacking (Fig. 1a), and hexagonal close-packed (hcp), with *ABAB* stacking. Colloidal hard-sphere crystals, however, are expected to form random stacking (rhcp), as the free energy difference between the fcc and hcp structures is small[36], and indeed rhcp structures have been observed when crystal nucleation from the liquid is homogeneous[37,38]. However, when crystals are nucleated on a flat substrate, scattering experiments[37] and simulations[39] have shown that they grow predominantly with a fcc structure, as observed here (Fig. 1b, green). The occasional hcp stacking (Fig. 1b, orange) form stacking faults in the fcc structure. These confocal images account for the nature of this behavior: the sample is composed of multiple fcc grains, which occur because nucleation takes place at multiple positions on the substrate simultaneously. The grains have a columnar shape and grow by stacking of hexagonal layers, as demonstrated in Fig. 1b by two snapshots in time of a growing crystal, where the grain boundaries are marked by the gray particles (see also Supplementary Movie 1).

Due to the particle buoyant weight, the increase of pressure along the thickness of the crystal results in its increasing compression, as demonstrated by the profiles of the volume-per-particle $v$ and the particle-particle distance in a plane parallel to the substrate $d_\parallel^0$ in Supplementary Fig. 1. Interestingly, we find that the crystal compression is isotropic so that the unit cell preserves its cubic shape (Methods). Therefore, the decrease of $v$ with $h$ is reflected by the decrease of $d_\parallel^0$, averaged over the thickness of the crystal, shown in Fig. 1d (top) by the blue symbols.

To impose strain, we grow crystals on templates with a square pattern[33,40] and spacing $d_t = 1.69(5)\,\mu m$, which is larger than the measured $d_\parallel^0(h)$, as marked by a dashed line in Fig. 1d (top). The templates constrain the first layer of particles and dictate growth of a single crystal along the [001] fcc direction in which particles follow the $A'B'A'B'$ stacking [Fig. 1a (bottom)]. During the early stages of growth, the in-plane particle-particle distance, $d_\parallel$, follows the template $d_\parallel = d_t$, as seen in Fig. 1d (top) and Supplementary Fig. 1. This also implies that, in contrast to unconstrained crystals, the compression with $h$ is not

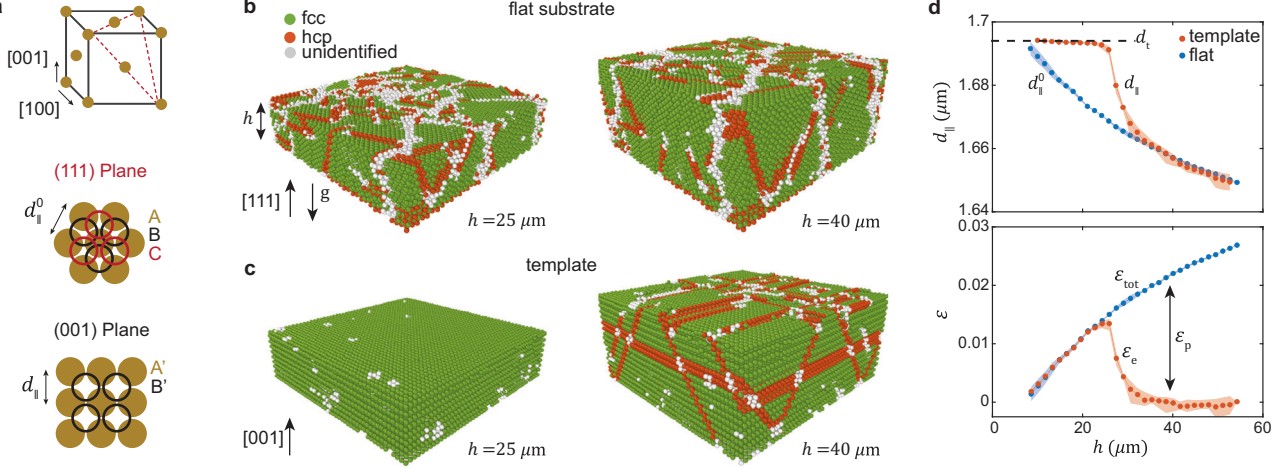

**Fig. 1 | Plastic relaxation and definition of strain measure. a** Particle stacking of {111} and {001} fcc planes. **b** Snapshots of a crystal growing on a flat substrate (unconstrained) show growth of columnar grains along the [111] direction. The crystal height and the direction of gravity are denoted by $h$ and $g$, respectively. Grains are predominantly fcc (green) with hcp stacking faults (orange). Grain boundaries are marked by gray particles that are not identified with a crystalline structure. **c** Snapshots of a crystal growing on a [001] template. At $h = 25\,\mu m$ (**c**, left) the crystal is defect free. By the time the crystal reaches $h = 40\,\mu m$ (**c**, right) hcp stacking faults have formed. **d**, top Evolution of $d_\parallel^0$ and $d_\parallel$, particle-particle

distances in a plane parallel to the substrate, averaged over the crystal thickness, for crystals grown on a flat substrate and on a template, respectively (see panel **a**). The onset of relaxation is marked by a sharp decrease of $d_\parallel$ as $h$ reaches a critical thickness $h_c \approx 26\,\mu m$. **d**, bottom Total ($\varepsilon_{tot}$) and elastic ($\varepsilon_e$) strains in constrained crystals, as defined in the main text. For $h < h_c$ imposed strains are accommodated elastically, $\varepsilon_{tot} = \varepsilon_e$, whereas for $h > h_c$, relaxation is mediated by plastic strain, $\varepsilon_p$. The shadows denote the variation over five distinct observation regions of a growing crystal. Here, the ionic strength $I = 2\,mM$ and template spacing $d_t = 1.69(5)\,\mu m$.

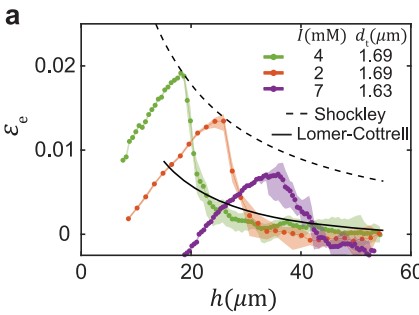

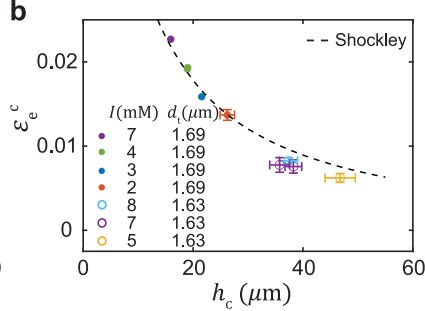

**Fig. 2 | Measuring and modeling plastic relaxation in colloidal crystals.**
**a** Evolution of the elastic strains $\varepsilon_e$ in growing crystals with different mismatch with the template. When crystals reach a critical thickness $h_c$ and elastic strain $\varepsilon_e^c$, which mark the onset of instability, sharp relaxation of $\varepsilon_e$ takes place. Modeling, based on Eq. (2), of relaxation by Shockley (dashed line) and Lomer-Cottrell (solid line) dislocations, captures the critical and residual strains, respectively. **b** Critical boundary $\varepsilon_e^c(h_c)$ compiled from experiments with different mismatch levels and predicted by the model; thinner crystals can sustain significantly larger elastic strains. **a**, **b** Legends indicate ionic strength, $I$ (mM), and template spacing, $d_t$. Error bars denote the variation over five distinct observation regions of a growing crystal. The orange example corresponds to the experiment analyzed in Fig. 1.

isotropic but is accommodated by contraction of the unit cell just in the direction parallel to gravity. As crystals reach a critical height $h_c \approx 26\,\mu m$, which marks the onset of instability, $d_\parallel$ begins to relax to the values measured for the unconstrained crystals grown on a flat substrate $d_\parallel^0$ [Fig. 1d (top) and Supplementary Fig. 1]. The perfect fcc structure we find for $h < h_c$ [Fig. 1c (left)] is no longer stable; the relaxation process is accompanied by formation of multiple hcp stacking faults [Fig. 1c (right) and Supplementary Movie 2].

We define the strain of constrained crystals, by using the unconstrained crystal as the reference frame, in analogy with the definition of the deviatoric strain. The total strains imposed by the templates in the [100] and [010] directions, which are equal, are given by $\varepsilon_{tot} = (d_t - d_\parallel^0)/d_\parallel^0$ and increase with $h$, as shown by the blue symbols in Fig. 1d (bottom). Importantly, during crystal growth, the imposed strains result from contraction of the reference spacing $d_\parallel^0$ with $h$, and do not involve mechanical stretching of the template spacing $d_t$. Similarly, the elastic in-plane strains are given by the difference between the particle distance in the constrained and unconstrained crystals, $\varepsilon_e = (d_\parallel - d_\parallel^0)/d_\parallel^0$ and reveal two regimes of deformation. Whereas for $h < h_c$, elastic strains account entirely for the imposed strains, $\varepsilon_e = \varepsilon_{tot}$, as crystals grow above $h_c$, $\varepsilon_e$ is relaxed, as shown by the orange symbols in Fig. 1d (bottom). This relaxation, and any subsequent increase of $\varepsilon_t$ with $h$, is accommodated by plastic deformation of the crystal, $\varepsilon_p = \varepsilon_{tot} - \varepsilon_e$.

We can address the relaxation process systematically by varying the imposed lattice mismatch. We take advantage of the nearly-hard sphere nature of our suspensions and decrease the electrostatic repulsion between the charged silica particles by adding NaCl and increasing the ionic strength, $I$. We find that, for the range $2 < I < 8$ mM, despite the relative softness of the inter-particle potentials, crystals are well described by the hard-sphere equation of state, with effective particle diameters $1.6 < 2R_{eff} < 1.63\,\mu m$ [Supplementary Fig. 2]. Therefore, we increase the mismatch with the template, $d_t = 1.69\,\mu m$, by increasing $I$ and decreasing the effective size of the particles. Crystals with a higher values of $I$ and higher mismatch reach higher $\varepsilon_e$ values and begin to relax at lower values of $h_c$, as can be seen by comparing the green ($I = 4$ mM) and orange ($I = 2$ mM) examples in Fig. 2a. To decrease the lattice mismatch we grow crystals on templates with a smaller spacing, $d_t = 1.63\,\mu m$. In this case, crystals reach even lower $\varepsilon_e$ values and begin the relaxation process at even higher values of $h_c$, as shown by the purple example in Fig. 2a. These three examples demonstrate that thinner crystals are stronger: critical elastic strains, $\varepsilon_e^c$, that mark the onset of instability, are higher in thinner crystals. This observation is systematically addressed in Fig. 2b, where the results of multiple experiments are compiled and values of $\varepsilon_e^c(h_c)$ are plotted. Remarkably, thin colloidal crystals can be strained elastically as much as 2%.

What is the underlying mechanism for relaxation? Plastic deformation in crystals is mediated by dislocations. To examine this process in detail we extract dislocation lines and their Burgers vectors, **b**, from the measured particle positions[41]. We identify four Shockley partial dislocations, one on each of the four {111} planes, with the Burgers vectors of the type $\mathbf{b} = \frac{1}{6}\langle 112 \rangle$[1], as illustrated in a diagram in Fig. 3a, where, for simplicity, only two out of the four planes are presented. Shockley dislocations are nucleated within the bulk of the crystal by forming closed dislocation loops (green lines) that bound hcp stacking faults (red particles), as shown in Fig. 3b (middle). The dislocation loops expand and reach the top and bottom surfaces of the crystal. Whereas the top surface is free and allows dislocations to escape from the crystal, the bottom surface is stiff and repels dislocations. Dislocation lines, therefore, have a typical shape: a long segment of edge character, parallel to and slightly above the template, called a misfit dislocation, which bends and forms screw-like segments (threads) that extend across the crystal thickness (Fig. 3a). As the relaxation process proceeds, the length of the misfit dislocations grows by glide of the dislocation threads [Fig. 3b (bottom)].

Remarkably, we find that (attractive) interactions between dislocations play a significant role in the relaxation process; two Shockley dislocations on separate {111} planes combine along the intersection line of the planes and form an (inverted) Lomer-Cottrell dislocation[1] with the Burgers vector $\frac{1}{6}[112] + \frac{1}{6}[11\bar{2}] \rightarrow \frac{1}{3}[110]$. The formation of Lomer-Cottrell dislocations is a two-step process: First, a Shockley dislocation is formed [Fig. 3c (top)], whereupon a second dislocation loop emerges in the vicinity of the first [Fig. 3c (middle)] and then expands and combines into a Lomer-Cottrell dislocation [red line in Fig. 3c (bottom)] in a zipping-like process. Although Lomer-Cottrell dislocations are immobile and are not allowed to glide on either of the {111} planes, they can extend by the motion of the Shockley threads. Lomer-Cottrell dislocations play an important role in strain-hardening of metals[1], and their formation in colloidal crystals have not been documented before. We show next that their ability to efficiently release misfit strain has major implications for the relaxation process.

We define a network of misfit dislocations by excluding the thread part of dislocations, which extends across the crystal height, and take into account only the misfit part of dislocations, which extends parallel to the interface, as defined in Fig. 3a. These networks consist of two perpendicular sets of parallel lines along the two principal directions of the template, [110] and [1$\bar{1}$0], as shown in Fig. 4a, where the network is projected along the thickness of the crystal. The dislocation networks consist of both Shockley and Lomer-Cottrell dislocations, although after initial rapid growth of the networks, the Lomer-Cottrell type dominates (Supplementary Fig. 3). The growth of the network occurs by nucleation of new

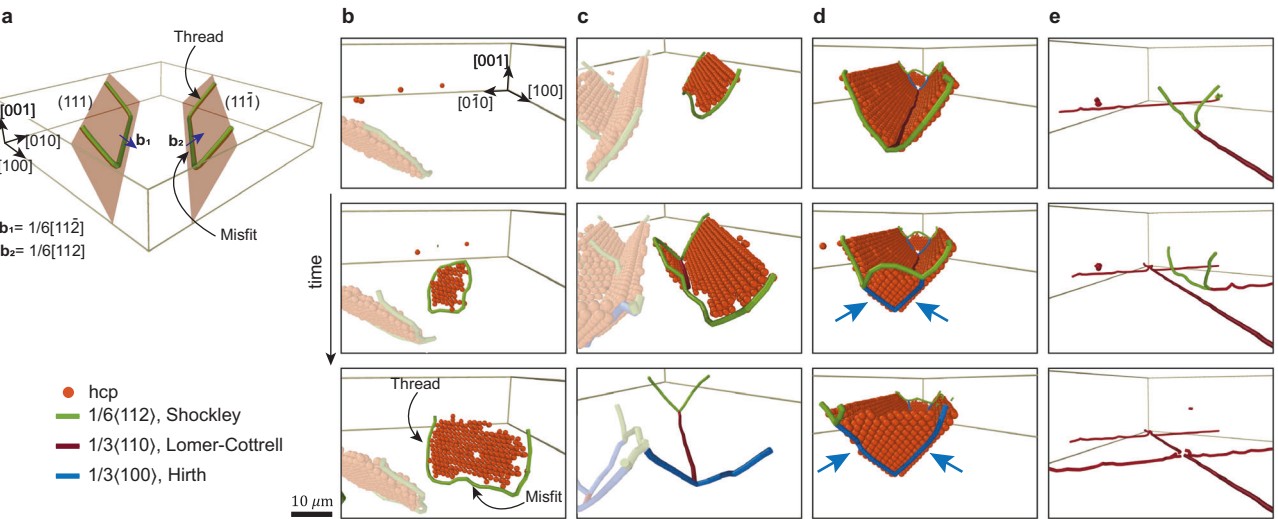

**Fig. 3 | Direct visualization of the nucleation of dislocations and their interactions. a** Schematic diagram of the Shockley dislocations that relax the tensile strain along the [110] direction. Misfit segments of edge character run along the [1$\bar{1}$0] direction, bend, and thread across the thickness of the sample. The two Shockley dislocations along [110] direction are omitted for clarity. **b–e** Time series ordered from top to bottom. Only hcp particles stacking faults and dislocations are presented. **b** Shockley dislocations (green), marking the boundaries of the stacking faults, nucleate by forming closed dislocation loops, and expand by glide of the thread segments. **c** A Lomer-Cottrell (LC) segment (red), $\frac{1}{3}$[110], is formed by

nucleation of a $\frac{1}{6}$[11$\bar{2}$] loop in the vicinity of a pre-existing $\frac{1}{6}$[112] misfit segment. Elongation of LC segments takes place by the glide of the Shockley threads. See definitions in (**a**). **d** Shockley thread segments of a LC dislocation react to form two Hirth thread segments (blue). As Hirth dislocations are immobile, the expansion of the Lomer-Cottrell segment is blocked. **e** Glide of the threads is blocked by repulsion from perpendicular misfit segments (middle) as suggested by the bending of the two threads. As the strain accumulates, crossing is observed (bottom). The apparent discontinuity of the segments at the crossing points (bottom) is due to a failure of the dislocation detection algorithm.

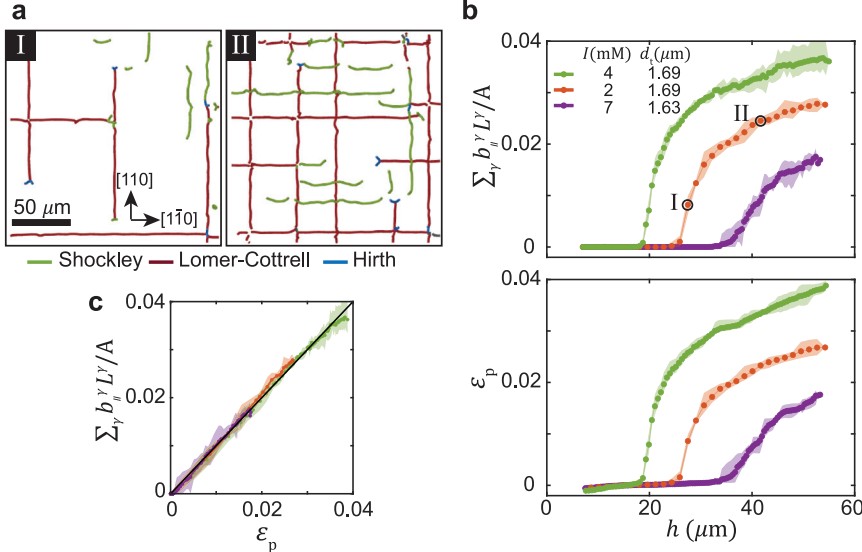

**Fig. 4 | Plastic relaxation by growth of dislocation networks. a** Snapshots of a growing misfit dislocation network, obtained by excluding dislocation threads, are plotted for different values of $h$, marked in (**b**). Colors correspond to different types of dislocations specified in the legend. **b** Plastic strain profiles $\varepsilon_p(h)$ obtained by either using Eq. (1) (top) or by a direct measurement $\varepsilon_p = \varepsilon_{tot} - \varepsilon_e$ (bottom).

**c** When predictions of Eq. (1) are plotted against the measured $\varepsilon_p$, all profiles collapse to a single line with a slope of one. **b, c** Legends indicate ionic strength, $I$ (mM), and template spacing, $d_t$. The shadows denote the variation over five distinct observation regions of a growing crystal.

dislocations and extension of existing ones, as can be seen by comparing the left and right panels in Fig. 4a.

Motion of a single dislocation results in a displacement **b** of particles across the slip plane. The average plastic strain associated with $N$ dislocations in a volume $V$, each spanning a slipped area **S**, is given by $\varepsilon_{ij} = \frac{1}{2}V^{-1}\sum_{k}^{N}(b_i^k S_j^k + b_j^k S_i^k)$[1]. In particular, the misfit strains in the two principal directions of the template, [110] and [1$\bar{1}$0], which are equal in our experiments, are mediated by two sets of parallel dislocations that grow along the [1$\bar{1}$0] and [110] directions, respectively (Fig. 4a). The

plastic strain, which relaxes the elastic misfit strain, therefore, can be written as

$$\varepsilon_p = \sum_{\gamma = S, LC} b_{\parallel}^{\gamma} L^{\gamma}/A \qquad (1)$$

Here, the sum over $\gamma$ refers to Shockley (S) and Lomer-Cottrell (LC) dislocations and $b_{\parallel}^{S} = a/\sqrt{3}$ and $b_{\parallel}^{LC} = 2a/\sqrt{3}$ are the in-plane components of their Burgers vectors, where $a$ is the fcc lattice constant. $L^{\gamma}$ is

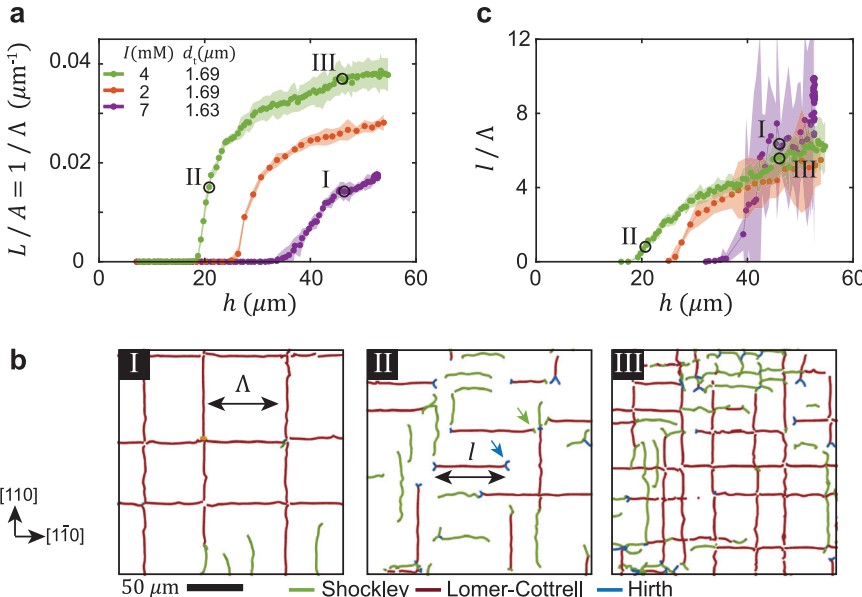

**Fig. 5 | Images and measures of misfit dislocation networks. a** Evolution of the areal density of dislocations $L/A = 1/\Lambda$ during growth of crystals with different degrees of lattice mismatch. **b** Three examples of dislocation networks marked in (**a**) and (**c**). The average dislocation spacing $\Lambda = (L/A)^{-1}$ and segment length $l$ are illustrated by double arrows in examples I and II, respectively. Examples of Hirth sessile threads (blue) and blocking by perpendicular segments are indicated by blue and green arrows, respectively. Note that the apparent discontinuity of the segments at the crossing points in examples I and III is due to a failure of the dislocation detection algorithm. **c** Evolution of $l/\Lambda$ with $h$. See legend in (**a**). **a**–**c** Examples I and II demonstrate networks with distinct structure, despite an identical values of $L/A$. In contrast, examples I and III show crystals of the same thickness. Their networks are similar: both are characterized by $l/\Lambda \approx 6$, despite a significant difference in areal dislocation density. The time evolution of the networks is shown in Supplementary Movies 3 and 4. The shadows in (**a**) and (**c**) denote the variation over five distinct observation regions of a growing crystal.

the total length of all the misfit segments in an area $A$, so that $L^\gamma/A$ is the areal density of each dislocation type.

Colloidal crystals provide a unique opportunity to examine directly the classical prediction of Eq. (1), as they allow direct measurement of strains and visualization of dislocations. We obtain $\varepsilon_p(h)$ profiles for the three typical experiments considered in Fig. 2a by either a direct measurement $\varepsilon_p = \varepsilon_{tot} - \varepsilon_e$ [Fig. 4b (bottom)] or by using Eq. (1) and summing over $L$ [Fig. 4b (top)]. When the predictions of Eq. (1) are plotted with respect to the measured $\varepsilon_p$, all profiles collapse to a single curve with a slope of one. This excellent agreement confirms that the plastic strains are mediated entirely by the growth of the network of misfit dislocations.

Although Eq. (1) establishes the connection between the length of the misfit network and $\varepsilon_p$, modeling the development of these networks poses significant challenges. To gain insight into this process we address the relaxation mediated by the Shockley and by the Lomer-Cottrell dislocations separately. The total elastic energy per unit area, for each type of dislocations, is approximated, following the classical misfit theory for isotropic linear elastic materials[5], by the sum

$$U^\gamma = \int_0^h \mathcal{E}_{el}^\gamma dz + L^\gamma/A \int \int \mathcal{E}_L^\gamma dx dz \qquad (2)$$

where, again, $\gamma$ refers to either S or LC. Here, $\mathcal{E}_{el}^\gamma \sim \mu(\varepsilon_{tot} - b_\parallel^\gamma L^\gamma/A)^2$ is the bulk elastic energy density, with $\mu$ the shear modulus. The energy density $\mathcal{E}_L^\gamma$ is associated with the energetic cost of introducing dislocations, where we ignore interactions among them, and integrated over $xz$ plane, perpendicular to the dislocation line (Methods).

The misfit theory is adapted here to model relaxation in colloidal crystals: We include in the model the height-dependence of $\varepsilon_{tot}(z)$ and $\mu(z)$ due to the change of the osmotic pressure with $z$[40], which results from the mass density mismatch between the particles and the solvent (Methods). We also account for the stiff substrate[42,43] by including in $\mathcal{E}_L^\gamma$ the finite distance of dislocations and their repulsion from the bottom

interface[13] (Supplementary Figs. 4 and 5). We expect, however, that our calculations provide a slight over estimate of $\mathcal{E}_L^\gamma$, as particles are free to move slightly inside the wells of the template, so that the idealized no-slip boundary conditions we assume here are not fully satisfied. To account for that, we introduce an adjustable parameter $\alpha$, so that $\mathcal{E}_L^\gamma \to \alpha \mathcal{E}_L^\gamma$ (Methods).

The two terms in Eq. (2) are affected by $L^\gamma$ in opposite ways; thus, equilibrium values of $L^\gamma/A$, and therefore $\varepsilon_e$, are determined by a minimization of the energy functional $\partial U^\gamma/\partial L^\gamma = 0$. The resulting $\varepsilon_e(h)$ calculated for Shockley ($\gamma = S$) dislocations agree with the onset of relaxation for $\alpha = 0.8$, as shown by the dashed lines in Fig. 2a and b. Remarkably, for the same $\alpha$, $\varepsilon_e(h)$ calculated for Lomer-Cottrell ($\gamma = LC$) dislocations describes the low residual values of $\varepsilon_e$. This analysis reveals the nature of the instability: the onset of relaxation is set by the high $\varepsilon_e$ required to nucleate and grow Shockley dislocations, whereas the sharp relaxation to the low $\varepsilon_e$ values is facilitated by the pair-wise interactions which allow formation of much more effective Lomer-Cottrell dislocations.

We now address the structure of the dislocation network. We start by defining the areal density of dislocations, $L/A$, where, for simplicity, we do not distinguish between the two types of dislocations, and take the total dislocation length to be $L = L^S + L^{LC}$. Crystals grown on templates with a higher mismatch are associated with higher values of $L/A$. Notably, our measurements demonstrate that the values of $L/A$ do not provide a unique definition of the network, as demonstrated in Fig. 5b by examples I and II; although both networks have similar densities (Fig. 5a), they show a very different structure. Whereas thicker crystals with a lower lattice mismatch (I) are characterized by well ordered arrays of long dislocation segments, thinner crystals with higher lattice mismatch (II) show a very fragmented network that consist of short segments.

To quantify the different structures of the dislocation networks, we consider two length scales. The average spacing between dislocations is defined by $\Lambda = (L/A)^{-1}$, as illustrated in Fig. 5b (left). It can be

seen simply that this applies to an ordered array of infinitely long parallel dislocations. The average length of dislocation segments is defined by $l = L/(N_s)$, where $N_s$ is the number of dislocation segments obtained by counting the average number of dislocation end-points [see illustration in Fig. 5b (middle)]. To characterize dislocation networks, we suggest a non-dimensional order parameter, $l/\Lambda$. Fragmented dislocations networks (example II) are characterized by low values of the order parameter, $l/\Lambda \sim 1$, whereas dislocation networks composed of ordered long arrays (example I) are characterized by high values, $l/\Lambda > 1$. Interestingly, we find that $l/\Lambda$ increases with $h$; a fragmented structure evolves into a rather ordered structure of long segments (Fig. 5c). This observation is demonstrated by examples II and III in Fig. 5b that show snapshots of the network in a growing crystal with a large lattice mismatch. Remarkably, the structure of dislocation networks in crystals grown to the same $h$, but with a different lattice mismatch, is similar; dislocation networks in examples I and III are characterized by a similar value of $l/\Lambda$, despite a significant difference in $L/\Lambda$.

We find two types of dislocation interactions that are critical to the evolution of $l/\Lambda$. Growth of Lomer-Cottrell segments is often blocked by the formation of Hirth sessile junctions[1]. The two Shockley threads, $\frac{1}{6}[112]$ and $\frac{1}{6}[11\bar{2}]$, at one end of a Lomer-Cottrell segment [Fig. 3d (top)], are converted into two Hirth threads, $\frac{1}{3}[100]$ and $\frac{1}{3}[0\bar{1}0]$, respectively, by reacting with a third Shockley dislocation, $\frac{1}{6}[11\bar{2}]$, which is nucleated near dislocation vertex at the bottom [Fig. 3d (middle)] and glides towards the top free surface of the crystal [Fig. 3d (bottom)]. The Lomer-Cottrell segment is blocked and is no longer able to expand by the glide of the two threads, as Hirth dislocations are immobile. Multiple segments blocked by the Hirth dislocation threads can be clearly identified by the blue v-shaped end points in networks with a fragmented structure, as indicated by a blue arrow in example II in Fig. 5b. Alternatively, our measurements show that expansion of dislocations can also be blocked by interaction with perpendicular misfit segments, as we demonstrate in the middle panel of Fig. 3e; two Shockley threads of a Lomer-Cottrell dislocation are repelled by a perpendicular segment, as suggested by the evident upward bending of the two threads. These measurements provide the first direct 3D visualization of their blocking mechanism, and confirm previous theoretical analysis[9,12,13]. An example of this blocking mechanism is highlighted by a green arrow in example II in Fig. 5b.

Interestingly, dislocations can overcome repulsion by perpendicular segments, as shown by the bottom panel in Fig. 3e, and also unzip the formed Hirth dislocation threads (Supplementary Movie 4). The measured increase of $l/\Lambda$ with $h$ (Fig. 5c) implies, therefore, that the strength of both blocking mechanisms should decrease with increasing thickness. Whereas the analysis of Hirth junctions in thin films has not been considered, previous theoretical studies[9,12,13] indeed predict that blocking by perpendicular segments is less effective in thicker crystals.

## Discussion

We have demonstrated the role of dislocation interactions in the plastic relaxation of strained colloidal crystals. Our analysis shows that these interaction mechanisms can be readily transferred to atomic systems, despite some of the unique properties of colloidal crystals, such as the relatively simple hard-sphere interparticle interactions and height-dependent osmotic pressure (see Modeling strain relaxation section in the Methods). These interaction mechanisms are particularly relevant to systems that satisfy several conditions: First, plastic strain should be mediated by Shockley partial rather than perfect dislocations. Dissociation of perfect into partial dislocations[1] in hard-sphere colloidal crystals[33,34,44–46] is energetically favorable due to vanishing stacking fault energy[36]. Partial dislocations, however, are not unique to colloidal crystals and are common to low-stacking fault metallic and semiconductor films[47]. Second, we have considered here

tensile misfit strain, whereas we expect different relaxation mechanisms if the substrate imposes compression. Finally, our analysis builds on the equilibrium theoretical framework, suggesting that lattice resistance to dislocation nucleation and glide plays a secondary role and does not inhibit dislocation kinetics. Although in some atomic systems the lattice resistance is strong and can result in a metastable film growth well above the critical height, our results should apply when this kinetic barrier to dislocation motion is lowered by elevating the film's temperature.

The comprehensive description of the relaxation process we provide here not only highlights the fundamental connection between dislocation interactions and the relaxation process, but also has practical implications. For example, accounting for the Shockley to Lomer-Cottrell transition that results in lower-than-expected residual strain should help improve the design of epitaxial semiconductor layers, as strain directly affects their band-gap. Furthermore, independent control of the spacing and length of the dislocation segments by misfit strain and film thickness can be utilized for defect engineering: longer segments are beneficial when dislocations are used as an easy path for the lateral transport of charges or dopants; the formation of fragmented networks with a high density of dislocation threads can be helpful surface seeds for the growth of novel micro- and nanostructures[17,24]. Finally, dislocations also play an important role in colloidal self-assembly[48]. As assembly of colloidal crystals is a primary route for creating photonic band-gap materials[20], the demonstrated control of dislocations could potentially be used to add different functionalities[49], such as creating optical propagation channels.

## Methods

### Sample preparation, crystal growth and imaging

We use silica particles (Micromod, Sicastar) with a diameter $2R = 1.55\,\mu m$ dispersed in a mixture of 64/36 (% by volume) of dimethyl sulfoxide (DMSO) and water that matches the index of refraction of the particles ($n = 1.43$). The gravitational height is $l_g = k_B T/\Delta \rho g v_p = 0.23\,\mu m$, where $v_p$ is the particle volume and $\Delta \rho$ is the difference between the particle and fluid densities. We add to the fluid $0.66 - 1\,mM$ Fluorescein-NaOH dye, to allow fluorescence imaging and $0 - 5\,mM$ NaCl to further decrease the Debye screening length. We avoid using higher concentrations of NaCl, as the particles begin to aggregate. Importantly, to obtain reproducible results, the particles are washed into the fluid mixture for several consecutive cycles of centrifugation and exchange of the supernatant. The concentration of ions in the solution is characterized by the ionic strength, $I$, defined by $I = 1/2 \sum c_i z_i^2$, where the sum includes both anions and cations, $c_i$ is the molar concentration, and $z_i$ is the charge number. For Fluorescein-NaOH, assuming full dissociation, $I = 3c_{Fluo}$, and for NaCl, $I = c_{NaCl}$, which gives $2 < I < 8\,mM$ for the concentrations we use.

Crystals are grown by sedimentation of the particles onto #1.5 coverslips with a 5 mm × 5 mm square pattern of ~500 nm deep wells with 1.63(5) μm or 1.69(5) μm spacing, fabricated by photolithography and reactive ion etching. These templated coverslips are glued to a 316 stainless steel cylindrical sample cell, 10 mm in diameter and 7 mm in height. To allow slower sedimentation, typically of ~5 crystalline layers per hour, we double the cell height by sample extension made of polyethylene. We find no evidence for rate-dependence of the relaxation process if the crystal growth rate is doubled. Furthermore, no dislocation nucleation is observed if the final height of the grown crystals is kept slightly below the critical value. In fact, if dislocations are induced in crystals below the critical height, for example, by increasing the laser power of the confocal, they quickly disappear once the laser power is reduced. We also note that due to the large thickness of the fluid, a temperature difference of ~1 °C between the objective and the sample cell is sufficient to generate slow convection currents in the fluid, which, over the course of the experiment, perturb the

sedimentation of the particles and the growth of the crystals. We find that by heating the sample cell by ~1 °C, these currents are prevented.

Particles are imaged with a Yokogawa CSU-W1 spinning-disk confocal scanner with 25 μm size pinhole and Zyla 4.2 sCMOS camera, set up to 60 ms exposure. The confocal module is attached to an inverted Leica microscope body (DMi8) equipped with 63 × /1.3 Glycerol objective lens, providing a field of view of 200 μm × 200 μm with negligible distortions. To avoid deterioration of the image quality during the time of the crystal growth (~10 h), we substitute Type-G immersion liquid ($n = 1.45$) with a Silicone-based fluid, Gelest Alt-143 ($n = 1.445$). The objective lens is mounted on a piezo objective scanner (PI, P-725), which allows us to image a 65 μm thick volume with 0.22 μm steps in less than 60 sec. We find that using high laser intensities can locally destroy the colloidal crystal. Furthermore, frequent acquisition even with mild laser intensities can facilitate formation of dislocations. We, therefore, minimize the laser intensity and find that limiting our imaging to every 7-15 minutes per field-of-view prevents these artifacts.

## Comparing sedimentation of constrained and unconstrained crystals

We compare here the growth process of crystals that are constrained by the templates with unconstrained crystals that are grown on flat substrates. We calculate the volume per particle (Voronoi volume, $v$) for all particles that are classified as either fcc or hcp[50] and obtain the profiles of $v(z)$ by averaging over the crystalline layers. Here $z$ is the coordinate along the gravity axis, where $z = 0$ denotes the substrate. $v(z)$ increases with $z$ as the crystal-fluid interface is approached, reflecting the decrease of osmotic pressure due to the decreasing weight of the particles, as demonstrated for constrained (orange) and unconstrained (blue) crystals in Supplementary Fig. 1a.

Although constrained and unconstrained crystals show identical $v(z)$ profiles, the corresponding profiles of the in-plane particle-particle distance, $d_\parallel(z)$, are different. The $d_\parallel(z)$ profiles measured during the growth of unconstrained crystals are an increasing function of $z$ and reflect the profiles of $v(z)$ [Supplementary Fig. 1(b, blue)]. In fact, we find that unconstrained crystals are compressed isotropically and the unit cell has a cubic shape, as the measured $d_\parallel$ profiles are related to the $v(z)$ profiles by $d_\parallel(z) = (\sqrt{2}v(z))^{1/3}$. Compression in the direction perpendicular to the gravity axis is not intuitive and reflects the behavior of fluids and not single crystals. This compression requires squeezing particles from the upper to the lower layers of the crystal. While this exchange of particles is impossible in perfectly ordered crystals, it can be accomplished through the grain boundaries or formation and annihilation of stacking faults. Non-trivial dynamics of the staking faults during the sedimentation process are indeed observed in Supplementary Movie 1. We will provide a detailed analysis of this process in a future publication.

In contrast, constrained crystals are not compressed isotropically. Below the critical height, $h < h_c$, $d_\parallel(z)$ follows closely the lattice spacing imposed by the template $d_\parallel(z) = d_t$, as demonstrated for $h = 22$ μm by the orange symbols in Supplementary Fig. 1b. Compression of the crystals, due to the weight of the particles, is accommodated only by the contraction of the unit cell in direction parallel to gravity. As the crystals grow above $h_c$, $d_\parallel(z)$ relaxes within the bulk of the crystal by roughly 3%, approaching the profile of the unconstrained crystal, as shown by an example for $h = 40$ μm in Supplementary Fig. 1b. Importantly, relaxation of $d_\parallel(z)$ leaves no signature in the profile of $v(z)$, and the tetragonal unit cell, therefore, reverts to the cubic shape measured for unconstrained crystals.

To quantify the relaxation process, we average the $d_\parallel(z)$ profiles over the crystal thickness. These averages, for both constrained and unconstrained crystals, are plotted for increasing crystal height in Fig. 1d of the main text.

## Effects of electrostatic repulsion on sedimentation profiles

Silica suspensions are stabilized by electrostatic repulsion between the particles, resulting from accumulated charge on the particle surfaces. The screening length is controlled by the amount of ions added to the fluid, which is quantified by the ionic strength of the solvent, $I$. The ionic strength affects the profiles of $v(z)$; the volume per-particle is smaller for higher values of $I$, as shown by three examples with different values of $I$ in Supplementary Fig. 2. The functional form of the $v(z)$ profiles, however, is not affected by the changes in $I$.

It is instructive to compare our measurements with a model for fully hard-sphere crystals. In general, the profiles of $v(z)$, or equivalently, profiles of the volume fraction, $\phi(z/l_g)$, are obtained by solving

$$\frac{d\Pi(z)}{dz} = -\frac{k_B T}{v_p} \frac{\phi}{l_g}, \tag{3}$$

if the density dependence of the osmotic pressure $\Pi(\phi)$ is known. For dispersions of hard-sphere colloidal particles, the equation-of-state is

$$\Pi(\phi) = \frac{k_B T}{v_p} \phi Z(\phi), \tag{4}$$

where $Z(\phi)$ is the compressibility factor given in ref. 51. Interestingly, we find that, despite the softness of the particle-particle potential, the hard-sphere model accurately describes the measured profiles $v(z) = v_p \phi^{-1}(z)$, if effective particle volumes $v_p$ are used, as shown by the dashed and solid black lines in Supplementary Fig. 2. The effective particle diameters $2R_{eff}$ that correspond to the inferred values of $v_p$ (legend in Supplementary Fig. 2) are 2% – 5% larger than the particle diameter, $2R = 1.55$ μm. We find that, for the different values of $I$, the surface-to-surface particle separation distance, $2R_{eff} - 2R$, is between 40 nm to 80 nm, which is more than ten times larger than the estimated Debye screening length. In this work we exploit this nearly-hard-sphere behavior of our suspensions to control the strains imposed on the crystals.

## Dislocation-substrate separation distance

Close examination of the 3D structure of dislocations presented in Fig. 3 of the main text and Supplementary Fig. 4a reveals a slight separation distance between the misfit dislocation segments and the templated substrate. To characterize this observation we measure the total length of misfit segments, which includes both Shockley and Lomer-Cottrell type dislocations, in thin slices of volume, defined by an area $A$ and width $dz = 0.5$ μm. The dislocation volumetric density, $L/Adz$, is plotted in the inset of Supplementary Fig. 5b as a function of $z$, coordinate along the crystal thickness [see coordinate system in Supplementary Fig. 5a]. Similar profiles are generated for different values of $h$, from which we obtain the average separation distance between the substrate and dislocation segments, $\delta$, as plotted in the main panel of Supplementary Fig. 5b. Interestingly, $\delta$ does not vary significantly during crystal growth. Considering the forces acting on the misfit dislocations[13], the balance between the repulsion from the substrate and the downward Peach-Koehler forces, that originate from the misfit strain, implies that $\delta$ should be an increasing function of $h$. This equilibrium condition, however, is not satisfied in our experiments as the majority of dislocations (~80%) are of the Lomer-Cottrell type (Supplementary Fig. 3); Lomer-Cottrell dislocations are sessile and can not adapt their $z$ position after their formation. We find, however, that $\delta$ varies between the different experiments and indeed increases with decreasing level of mismatch as summarized in Supplementary Fig. 5c, where the measured $\delta$ are plotted versus the critical thickness, $h_c$.

## Modeling strain relaxation

We consider strain relaxation in a thin film of an isotropic linear elastic material characterized by a shear modulus $\mu$ and a Poisson ratio $v$. As

the lateral dimensions of the film are much larger than its height and the length of dislocation segments, we ignore any lateral edge effects. The total elastic energy (per unit area) is approximated[5,6] by the sum of the bulk elastic energy and the energetic cost associated with introducing a network of non-interacting dislocations $U = U_{el} + U_L$:

$$U = \int_0^h \mathcal{E}_{el} dz + 2\frac{L}{A} \int \int_S \mathcal{E}_L dx dz \qquad (5)$$

Here, $L/A$ is the total length of dislocations per area, $\mathcal{E}_{el}$ and $\mathcal{E}_L$ are the corresponding energy densities, and the factor of 2 accounts for the two perpendicular arrays of dislocations, which, for simplicity, was omitted in the main text. The first term in Eq. (5) can be calculated by decomposing the bulk elastic strain $\varepsilon_e = \varepsilon_{tot} - \varepsilon_p$ into two parts: (1) $\varepsilon_{tot}$ is the imposed misfit strain, which results from the difference between the film and substrate lattice-spacing and (2) $\varepsilon_p = b_\parallel L/A$ is the strain relaxed by dislocations, where $b_\parallel$ is in-plane component of the Burgers vector. In this case, $\mathcal{E}_{el} = \chi\mu(\varepsilon_{tot} - b_\parallel L/A)^2$, where the numerical factor $\chi$ depends on the choice of the loading conditions, as will be addressed below. The second term in Eq. (5) is obtained by integrating $\mathcal{E}_L$ over an area $S$, perpendicular to the dislocation line, with outer and inner radii $h$ and $r_c$, respectively [see Supplementary Fig. 5a]. The latter, known as the dislocation core radius, is used to regularize the mathematical singularity along the dislocation line and is taken as $r_c = b/\beta$, with $\beta$ often considered to be of order unity[46].

In the simplest form of the classical thin film theory, the film and the substrate are assumed to have identical elastic moduli. In this case, dislocations are expected to form at the interface separating the film and the substrate, and the total energy is

$$U = \chi\mu(\varepsilon_{tot} - b_\parallel L/A)^2 h + 2\frac{L}{A}\frac{\mu b^2}{4\pi(1-\nu)}\log\left(\frac{h}{r_c}\right) \qquad (6)$$

where the second term is the energy (per area) of an edge dislocation. The elastic strain of the film is obtained by energy minimization $\partial U/\partial(L/A) = 0$:

$$\varepsilon_e = \varepsilon_{tot} - b_\parallel L/A = \frac{b^2/b_\parallel}{4\pi(1-\nu)\chi}\frac{\log(h/r_c)}{h} \qquad (7)$$

The critical height $h_c$ is determined by setting $L/A = 0$, and the increase of $h$ above $h_c$ results in the decrease of $\varepsilon_e$. Equation (7) demonstrates that Lomer-Cottrell dislocations $\mathbf{b} = a/3[1,1,0]$ are more efficient than Shockley dislocations $\mathbf{b} = a/6[1,1,2]$ in relaxing strain: the residual elastic strain $\varepsilon_e(h)$ relaxed by LC dislocations is lower, as $b^2/b_\parallel = a\sqrt{2}/3$ and $b^2/b_\parallel = a\sqrt{2}/2$ for LC and S, respectively.

We adapt the classical thin film theory to address the relaxation of colloidal samples by including in the model the height-dependence[40] of the osmotic pressure and particle density (Supplementary Fig. 2). The misfit strain is defined by $\varepsilon_{tot} = (d_t - d_\parallel^0)/d_\parallel^0$, the difference between the undeformed crystal lattice-spacing and template spacing $d_\parallel^0$ and $d_t$, respectively. The height-dependence of $\varepsilon_{tot}(z)$ is attributed to $d_\parallel^0(z)$. We take $d_\parallel^0$ to be the in-plane nearest neighbor distance $d_\parallel^0(z) = (\sqrt{2}\nu(z))^{1/3}$, where $\nu$ is the volume per particle. The shear modulus is given by $\mu(z) = \frac{3(1-2\nu)}{2(1+\nu)}K(z)$, where the bulk modulus profile $K(\phi) = -\phi^{-1}\partial\Pi/\partial\phi^{-1}$ is obtained from the $\phi(z)$ solution to Eq. (3) and Eq. (4). The calculated $\mu(z)$ profile is used to evaluate the integrals in Eq. (5). Here we neglect the height-dependence of $\nu$ and take $\nu = 0.24$[26].

We further account in our model for the infinite stiffness of the substrate. To obtain $\mathcal{E}_L$, we consider only interactions between dislocations and the template; interactions between dislocations and the top free surface and dislocation-dislocation interactions are neglected. The solution for an edge dislocation at a distance $\delta$ [Supplementary Fig. 5a] above an interface of perfectly bonded two semi-infinite materials is given in[42,43]. Here we assume no-slip boundary conditions

at the template and consider the limit of infinite elastic contrast between the two materials. The Airy-stress function, given explicitly by the solution, is numerically differentiated to obtain the two-dimensional stress tensor $\sigma_{ij}(x,z)$. We find that the strains inferred from this solution agree well with the strain fields measured in the vicinity of the Shockley and Lomer-Cottrell dislocation segments (Supplementary Fig. 4). $\mathcal{E}_L(x,z;\delta,\mu,\nu)$ is evaluated according to $\mathcal{E}_L = \frac{1}{2}\Sigma\sigma_{ij}\varepsilon_{ij}$ for $x,z \in S$ [Supplementary Fig. 5(a)]. $\mathcal{E}_L$, therefore, accounts for both the self-energies of dislocations and their interactions with their images. We expect, however, that our calculations provide a slight overestimate of $\mathcal{E}_L$, as particles are free to move slightly inside the wells of the template so that the idealized no-slip boundary conditions we assume here are not fully satisfied. To account for that, we introduce an adjustable parameter $\alpha$, so that $\mathcal{E}_L \rightarrow \alpha\mathcal{E}_L$.

Our measurements also show that crystals grown on templates and the reference crystals grown on flat substrates show very similar profiles of $\nu(z)$ [Supplementary Fig. 1]. This suggests that the deformation process takes place with the condition $\Sigma_i\varepsilon_{ii} = 0$. This constraint is important when energy densities $\mathcal{E}_{el}$ and $\mathcal{E}_L$ are calculated and whenever strains are transformed to stresses and vice versa. For example, for bi-axial elastic strain we obtain $\chi = 6$, which is in contrast with the classical plane stress ($\sigma_{zz} = 0$) theory in which $\chi = 2(1+\nu)/(1-\nu)$.

Finally, $\varepsilon_p$ is obtained by a variation of the energy functional (Eq. (5)), $\partial U/\partial(L/A) = 0$:

$$\varepsilon_p = \left(\int_0^h \mu(z)dz\right)^{-1}\left(\int_0^h \mu(z)\varepsilon_{tot}(z)dz - \frac{\alpha}{\chi b_\parallel}\int\int_S \mathcal{E}_L(x,z;\delta,\mu(z),\nu)dxdz\right) \qquad (8)$$

whereas the elastic strain is defined by $\varepsilon_e = \int_0^h \varepsilon_{tot}dz - \varepsilon_p$.

We are now in a position to compare the relaxation model with the measured strains. We use $2R_{eff} = 1.63\,\mu m$ and $d_t = 1.69\,\mu m$ to calculate the profiles of $\mu(z)$ and $\varepsilon_{tot}(z)$, and find that for the range of values used in the experiment the differences are insignificant. The functional form of $\delta(h)$ is obtained by linear regression of the measured values of $\delta$, as shown by a solid black line in Supplementary Fig. 5c. The only two adjustable parameters are $\alpha$ and $\beta$. We find that the relaxation model for Shockley dislocations accurately describes our measurements for $\alpha = 0.8$ and $\beta = 1$, as shown by a black dashed line in Supplementary Fig. 5d. Importantly, for the same values of $\alpha$ and $\beta$, the model for Lomer-Cottrell dislocations describes well the residual strains, as shown in Fig. 1a of the main text. The choice of $\beta = 1$ is consistent with the strain analysis in Supplementary Fig. 4d and with ref. 52.

The agreement between the model and the measured strains supports the idea that the sharp relaxation results from the formation of the more effective Lomer-Cottrell dislocations. We expect that this result is readily transferred to atomic systems, as a qualitatively similar scenario is suggested by Eq. (7).

## Data availability
The data is available on request.

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

## Acknowledgements

This work was supported primarily by the Harvard MRSEC program of the National Science Foundation under award number DMR 20-11754. I.S. acknowledges a fellowship from USIEF Fulbright program.

## Author contributions

I.S. and S.K. performed the experimental work and analysis. I.S., S.K., D.A.W. and F.S. contributed to the design and planning of the work, and to the writing of the manuscript.

## Competing interests

The authors declare no competing interests.
