## [Peer Review File · Nature Communications]

Dislocation interactions during plastic relaxation of epitaxial colloidal crystalsREVIEWER COMMENTS

Reviewer #1 (Remarks to the Author):

Manuscript "Dislocation interactions during plastic relaxation of epitaxial colloidal crystals" by Dr Spaepen et al.

I have read the manuscript with great interest. The authors studied partial dislocations in colloidal crystals grown on flat and patterned substrates. The defects in colloidal crystals are very important from two perspectives (or, at least two). One (1) is fundamental as colloidal crystal defects are somewhat similar to what one typically finds in metals. The "seeability" of colloids allows investigating the defects cores with "atomic" resolution in real-time. The other perspective (2) is the photonic applications of colloidal crystals, where defects can, on one hand, deteriorate the photonic properties and, on the other hand, can allow for the introduction of additional functionalities using controlled incorporation of desired defects and pre-programmed positions. In particular, partial dislocations reduce the local density of the particles in the core (in contrast to stacking faults). This could *potentially* be used to create light waveguides through the photonic material within the band gap. These opportunities of defect engineering could be mentioned to attract more attention to the topic.

I have several comments/suggestions for the authors of this interesting work.

Line 63, "fluid NaCl" sounds confusing. It is obviously not about melts of NaCl but about adding some salt to the suspension. As described in more detail in the SI, NaCl concentrations were in the range of 0-5 mM. Can the authors describe in more detail how the concentration was selected for each experiment?

I wonder at which height the in-plane d-spacing shown in Fig. 1D is measured (the caption says "...in a plane parallel to the substrate...")? Is it averaged over the whole crystal or within a specific plane?

Since the authors quantitatively looked at the crystal strain field, can they say what is the width of the (core of) partial dislocations? My experience is that while the core of Shockley dislocations is usually rather sharp, with a width comparable to the lattice period, Lomer-Cottrell dislocations are often broader. Can the authors confirm this (or not)?

The particles used in this work are pretty large and their mass density is not matched that of the solvent. As a result, their gravitational length is less than one-sixth of their diameter. This means that the osmotic pressure must change significantly over the crystal leading to the height dependence of the lattice parameter. The introduced strain is incompatible with the long-range periodic order. Can the authors comment on the effect of gravity on the results?

Reviewer #2 (Remarks to the Author):

This manuscript presents the study of dislocation dynamics in colloidal crystals. The method offers a unique advantage in that the particles can be observed directly using microscopy techniques, allowing for a detailed analysis of the behavior of dislocations in real-time. In molecular crystals, the study of dislocations is often limited by the fact that they are challenging to observe directly. The key finding in this paper is how networks of dislocations can interact to result in rapid plastic relaxation. Overall, these are interesting results that illustrate an important phenomenon in materials and advance the application of colloids as molecular models.

Some questions to clarify some of the text:

1) The formation of the crystal requires using a flat surface with the sedimentation of spheres

into a fcc order with hcc. which leads to stacking faults. Then the particle weight is used to compress the colloidal crystal. Why do the authors suggest that the crystal strain is isotropic?

2) The strain is associated with the crystal compression and compared to an unconstrained crystal. The type of dislocation, its energy, and its organization is presented to highlight the mechanism of crystal relaxation. The ionic strength is used to control the interparticle position. With the number of variables, It is difficult to identify which variables are key to the dislocation generation and plastic deformation.

3) A key result is the network Lomer-Cottrell dislocations which are a type of dislocation that form in crystalline materials with high stacking fault energy. Typically, they are a type of edge dislocation that are locked in place, but the authors show an evolution in the crystal and growth of dislocation as a function of crystal thickness. How do these dislocations break to accommodate the crystal strain?

4) What is the role of the areal size on crystal relaxation? The authors are normalizing the energy and strains over area, but it seems like the area should influence the dislocation generation. If the crystals are significantly larger (or smaller), will the ratio of the Lomer-Cottrell to Shockley dislocations change?

Reviewer #3 (Remarks to the Author):

Review of "Dislocation interactions during plastic relaxation of epitaxial colloidal crystals"

In this study, the authors grow colloidal crystals (comprised of spheres) on templates and observe the growth and interaction of dislocations that relax the strain due to the template. The authors identify a two-step mechanism that accompanies relaxation; the nucleation of Shockley dislocations and subsequent merging into a Lomer-Cottrell dislocation. They provide convincing evidence that the growth of Shockley dislocations is the limiting step and requires large strain. Overall, I find the work to be of good quality and I have only minor comments (see below). The study appears to be carefully done and reasonably interpreted.

However I am reluctant to recommend publication in the current state, as the study's significance is under-developed. I can explain what I mean through the following points:

1. The main motivation given for this work is the importance of dislocations to the properties of thin-film materials with electrical and magnetic properties. The authors claim that dislocation interactions in epitaxial materials are poorly understood due to the difficulty of observing them directly, and propose that colloidal materials can be used as an easier-to-observe model system. I find this justification somewhat weak. Questions about dislocation evolution can be split roughly into two categories: (A) Long-ranged interactions where the material can be treated as a continuum or (B) Close-ranged interactions where the dislocation core structure is important. I do not believe we can learn very much about (B) in atomic systems from colloidal models, as the details of the interactions between atoms can be important. My interpretation is that this study falls into category (A). However I am left unsure what has been added to our understanding, aside from confirmation of preexisting simulation and theoretical results. I do not see how an experimental observation in a 'model system' is significantly different from a theory or numerical result - all three are models describing atomic systems. This is an issue of framing - the results obtained by the authors

are interesting and significant for other researchers seeking to understand colloidal crystals, but are underwhelming as a demonstration of the ability of colloidal crystals to reveal new insights about atomic materials.

2. Most importantly, it is not clear to me reading this paper if the mechanism the authors describe (Shockley to Lomer-Cottrell misfit strain relaxation) is transferable to any atomic systems. What is the major cause of the trend of $\epsilon_e(h)$ shown in fig. 2A? Does the second term in eq. 2 contribute? At first glance it appears not to. Is this effect due then mostly to variations in $\epsilon_{tot}(z)$ and $\mu(z)$? This would seem to make the S to LC relaxation mechanism less general and maybe applicable only to colloid films under gravity. If this interpretation is incorrect I would urge the authors to discuss what atomic systems this result applies to.

3. In general, I find the consequences of this study under-explored. How can a researcher growing metallic or semiconductor thin films benefit from this work? With the current motivation of the paper, it seems like this is a logical and reasonable extension. On the colloidal side, where the effects are obviously applicable, how can they be exploited? While I am skeptical of colloids as atomic models, colloids do possess many interesting designable properties that aren't readily available with atoms. Examples include anisotropic shape, patchiness, or large responses to light/heat/solvent chemistry. Even something trivial, like density matching with the solvent - how might that impact the reported mechanisms? In this study, thin films are resistant to dislocation growth. A logical next question is, given that many properties of colloids can be tuned, how can the same resistance be extended to thicker (colloidal) films?

Minor scientific points:

1. Line ~84: The authors observe that crystal films grown on flat substrates have isotropic compression of their units cells deep within the layer due to the weight of particles above. Isotropic compression is quite strange here - this is an anisotropic loading condition and one would expect anisotropic compression. The authors acknowledge that this is counter-intuitive and offer the explanation that it is "most likely due to the exchange of particles through the grain boundaries". What does this mean? I don't understand how that explains the observation.

2. Line 129: The authors note that thin films can be strained to larger values before plastically relaxing compared to thick films. As far as I can tell, these films are grown linearly in time at a rate of 5 layers per hour. This means that thick films are somewhat older than thin films. Does this difference in total sample lifetime play a role here? Does the longer life of thick films allow for more time to nucleate dislocations?

Small notes:

1. Line 85: It should be stated in the main text that $d_{\langle\langle\rangle\rangle}(h)$ is an average over all planes in the film, i.e. an average of $d_{\langle\langle\rangle\rangle}(z)$. I found the wording confusing in the main text.

2. Line 105: The authors state that "tensile strains result from contraction of the reference spacing $d_{\langle\langle\rangle\rangle}^0$ with h , and do not involve any mechanical stretching of the sample". This is confusingly worded - strain is a mechanical stretching away from a reference. I think it would be more clear to say that $d_{\langle\langle\rangle\rangle}$ does not increase with h (if that is true).

3. Fig 3: It is difficult to quickly understand the orientation of the camera in panels B-E. Can coordinate arrows (as in A) be drawn in the corner of each image?

4. Fig 5: The segment length λ is difficult to understand from the label in panel B II. It is simply very hard to tell which of the dislocation crossings (for instance in panel B I) are endpoints. If none of the crossings are segment ends, then this should be stated. If some are, then maybe they can be marked on the plots somehow.

We thank the reviewers for their helpful comments. We have revised the manuscript and the Methods section to address the questions and comments that were raised in the reviews. As a result, we believe that the modified manuscript is both improved and clearer. Below we note both our detailed response together with the changes made to the manuscript to address each of the reviewers' comments.

Reviewer #1:

Manuscript "Dislocation interactions during plastic relaxation of epitaxial colloidal crystals" by Dr Spaepen et al.

I have read the manuscript with great interest. The authors studied partial dislocations in colloidal crystals grown on flat and patterned substrates. The defects in colloidal crystals are very important from two perspectives (or, at least two). One (1) is fundamental as colloidal crystal defects are somewhat similar to what one typically finds in metals. The "seeability" of colloids allows investigating the defects cores with "atomic" resolution in real-time. The other perspective (2) is the photonic applications of colloidal crystals, where defects can, on one hand, deteriorate the photonic properties and, on the other hand, can allow for the introduction of additional functionalities using controlled incorporation of desired defects and pre-programmed positions. In particular, partial dislocations reduce the local density of the particles in the core (in contrast to stacking faults). This could *potentially* be used to create light waveguides through the photonic material within the band gap. These opportunities of defect engineering could be mentioned to attract more attention to the topic.

Our response:

We are glad the reviewer finds our manuscript interesting, and we are grateful for motivating us to discuss the role of defects in colloidal photonic crystals. Following the reviewer's suggestion, we have modified the introduction to include the negative effects of dislocations on the photonic band gap of colloidal crystals:

Line 41- "Greater insight into thin film dislocation dynamics is also technologically important \cite{Tsao_book,Freund_book}: On the one hand, dislocations have detrimental effects such as decreasing electrical conductivity \cite{Chen2020}, photoconductivity \cite{Chen2020}, ferroelectricity \cite{Chu_2004}, thermoelectricity \cite{Zheng_2023}, and photonic band-gap \cite{Vlasov_2001}, while on the other hand, they have been used to advantage to control the functionality of thin films, enabling enhancement of superconductivity \cite{Dam1999,Llordes_2012}, switching of electrical resistance \cite{Szot_2006}, and fabrication of ordered nanostructures \cite{Brune_1998} and nanoscale ferromagnetic elements \cite{Sugiyama_2013}"

We now discuss the implications of our results at the end of the manuscript, which includes the new opportunities in the context of photonic crystals that control of dislocations allows:

Line 295- "Finally, dislocations also play an important role in colloidal self-assembly \cite{Li_2021}. As assembly of colloidal crystals is a primary route for creating photonic band-gap materials \cite{Vlasov_2001}, the demonstrated control of dislocations could potentially be used to add different functionalities \cite{Hilhorst_2013}, such as creating optical propagation channels."

Reviewer #1:

I have several comments/suggestions for the authors of this interesting work.

Line 63, “fluid NaCl” sounds confusing. It is obviously not about melts of NaCl but about adding some salt to the suspension. As described in more detail in the SI, NaCl concentrations were in the range of 0-5 mM. Can the authors describe in more detail how the concentration was selected for each experiment?

Our response:

The reviewer is correct, NaCl was dissolved in the solution. To avoid any confusion, we have modified the text to read:

Line 65 - “We disperse silica particles with a diameter $2R=1.55 \mu\text{m}$ in an index-matched fluid with Fluorescein–NaOH dye and control the Debye screening length of the particle solution by adding NaCl. ”

The concentration of NaCl was selected by trial and error. We identify the maximal concentration, $\sim 6\text{mM}$, above which the particles start to aggregate. We then use 0-5mM NaCl to span a broad range of strains (Fig. 2B). To clarify that, we modify the SI to read:

Line 425 - “We add to the fluid 0.66-1 mM Fluorescein–NaOH dye, to allow fluorescence imaging and 0-5 mM NaCl to further decrease the Debye screening length. We avoid using higher concentrations of NaCl, as the particles begin to aggregate.”

Reviewer #1:

I wonder at which height the in-plane d-spacing shown in Fig. 1D is measured (the caption says “...in a plane parallel to the substrate...”) ? Is it averaged over the whole crystal or within a specific plane?

Our response:

We thank the reviewer for pointing out this misunderstanding. The in-plane spacing and the strains in Fig. 1D were averaged over the crystal’s thickness. The corresponding depth-dependent profiles are discussed in the Methods and plotted in Fig. S1. Importantly, although we choose to simplify the main text by presenting the averages, our modeling of the relaxation process presented in the Methods fully accounts for these variations in depth.

To clarify this point, we modified the main text to:

Line 87 - “Due to the particle buoyant weight, the increase of pressure along the thickness of the crystal results in its increasing compression, as demonstrated by the profiles of the volume-per-particle v and the particle-particle distance in a plane parallel to the substrate d_{\parallel}^0 in Fig. S1. Interestingly, we find that the crystal compression is isotropic so that the unit cell preserves its cubic shape \cite{Methods}. Therefore, the decrease of v with h is reflected by the decrease of d_{\parallel}^0 , averaged over the thickness of the crystal, shown in Fig. 1D (top) by the blue symbols.”

The Fig. 1 caption now reads:

“(D, top) Evolution of d_{\parallel}^0 and d_{\parallel} , particle-particle distances in a plane parallel to the substrate, averaged over the crystal thickness, for crystals grown on a flat substrate and on a template, respectively. The onset of relaxation is marked by a sharp decrease of d_{\parallel} as h reaches a critical thickness $h_c \approx 26 \mu\text{m}$.”

Reviewer #1:

Since the authors quantitatively looked at the crystal strain field, can they say what is the width of the (core of) partial dislocations? My experience is that while the core of Shockley dislocations is usually rather sharp, with a width comparable to the lattice period, Lomer-Cottrell dislocations are often broader. Can the authors confirm this (or not)?

Our response:

Following the reviewer’s suggestion, we now provide a detailed analysis of the strain fields near Shockley and Lomer-Cottrell dislocations, which is summarized in the new Fig. S4. We find that the predictions of the singular linear elastic solution, which includes the effects of the stiff substrate, agree well with the measured strains. We find that the core width, if defined as the region of deviations from linear elasticity, is no wider than the Burgers vector. By this definition we don’t see a difference between two types of dislocations. We thank the reviewer for stimulating us to add this analysis to the paper.

“Fig. S4. Crystalline structure and elastic strains near Shockley (top row) and Lomer-Cottrell (bottom row) edge dislocations. (a) The dislocation lines run perpendicular to the page and marked by \otimes ; they bound hcp (orange) stacking faults in fcc (green) crystals. The crystalline structure is unidentified (white particles) in the close vicinity of the dislocation lines. The layer of the unidentified particles at the bottom marks the first layer of particles above the rigid template, which establishes zero displacement boundary conditions. (b) Color map of the resolved elastic strains ϵ_{bn} , where \vec{n} is the normal to the plane defined by \vec{b} and the dislocation line, as shown in (a). We subtract the value of ϵ_{bn} that corresponds to the average misfit strain. (a,b) The particle positions were averaged over 15 layers along the dislocation line ($[1\bar{1}0]$ direction). (c) Linear

elastic prediction of ε_{bn} for dislocations near an infinitely stiff substrate \cite{Dundurs_1965}, with no adjustable parameters. (d) The measured ε_{bn} (solid points) and the analytic solution (black lines) are compared along the dashed lines in panel (c)."

Line 604 – "Importantly, for the same values of α and β , the model for Lomer-Cottrell dislocations describes well the residual strains, as shown in Fig.1(a) of the main text. The choice of $\beta = 1$ is consistent with the strain analysis in Fig.S4(d) and with Ref.~\cite{Hilhorst_2011}."

Reviewer #1:

The particles used in this work are pretty large and their mass density is not matched that of the solvent. As a result, their gravitational length is less than one-sixth of their diameter. This means that the osmotic pressure must change significantly over the crystal leading to the height dependence of the lattice parameter. The introduced strain is incompatible with the long-range periodic order. Can the authors comment on the effect of gravity on the results?

Our response:

As the reviewer points out, gravity plays a crucial role in our experiments. On the one hand, gravity provides a simple method of confining the particles, which allows us to grow colloidal single crystals with a varying degree of imposed misfit strain. On the other hand, gravity indeed sets a certain level of complexity in modeling the relaxation process that is not present in atomic systems due to the variation of the lattice constant along the height of the crystal.

The depth-dependent compression of crystals is demonstrated by the volume-per-particle $v(z)$ profiles in Fig. S1(a) and reproduced below. Notably, the profiles measured during the growth of constrained crystals (crystals grown on a template) and unconstrained crystals (crystals grown on a flat substrate) are identical. Interestingly, in contrast to the $v(z)$ profiles, the in-plane particle-particle distance $d_{||}$ profiles of the two types are different. We find that unconstrained crystals (blue profiles) show isotropic compression: $d_{||}(z)$ profiles reflect the profiles of $v(z)$, implying that the unit cell preserves its cubic shape. The long-range order, as the reviewer suggests, is destroyed by the presence of grain boundaries and stacking faults as shown in Fig. 1B. In contrast, constrained crystals (orange profiles) are not compressed isotopically: Prior to the relaxation, $d_{||}(z)$ profiles follow the template spacing d_t (Fig. S1b, $h = 22\mu\text{m}$), which implies that compression is accommodated in the z direction resulting in a tetragonal unit cell. After the nucleation of dislocations, $d_{||}(z)$ profiles relax to the functional form measured in unconstrained crystals (Fig. S1b, $h = 40\mu\text{m}$).

Fig. S 1

We consider the depth dependence of the lattice spacing in our modeling of the relaxation process. First, we use the $d_{||}(z)$ profiles measured during the growth of unconstrained crystals as a reference frame for defining the strain of constrained crystals $\varepsilon(z)$. In this perspective, although prior to the relaxation the spacing is constant $d_{||}(z) = d_t$, strain varies along the height of the crystal and increases as the template is approached. Second, we account for depth-dependent compression by relating the $v(z)$ profiles to the depth dependence of the shear modulus $\mu(z)$. Both $\varepsilon(z)$ and $\mu(z)$ profiles allow us to integrate the elastic energies in Eq. 2 and quantitatively account for the relaxation process. While the effects of gravity do require some detailed attention, the presence of gravity does not change any of our conclusions on the role of dislocation interactions during epitaxial growth.

We appreciate the reviewer's interest in the details of our analysis. We now explicitly mention the density mismatch and refer the reader to the Methods for more detailed discussion of the effects of gravity:

Line 200 - "The misfit theory is adapted here to model relaxation in colloidal crystals: We include in the model the height-dependence of $\varepsilon_{\text{tot}}(z)$ and $\mu(z)$ due to the change of the osmotic pressure with z \cite{Jensen_2013}, which results from the mass density mismatch between the particles and the solvent \cite{Methods}. We also account for the ..."

Reviewer #2 (Remarks to the Author):

This manuscript presents the study of dislocation dynamics in colloidal crystals. The method offers a unique advantage in that the particles can be observed directly using microscopy techniques, allowing for a detailed analysis of the behavior of dislocations in real-time. In molecular crystals, the study of dislocations is often limited by the fact that they are challenging to observe directly. The key finding in this paper is how networks of dislocations can interact to result in rapid plastic relaxation. Overall, these are interesting results that illustrate an important phenomenon in materials and advance the application of colloids as molecular models.

Our response:

We thank the reviewer for the accurate and concise summary of our results and are glad that the reviewer finds our results interesting.

Reviewer #2:

Some questions to clarify some of the text:

1) The formation of the crystal requires using a flat surface with the sedimentation of spheres into a fcc order with hcc. which leads to stacking faults. Then the particle weight is used to compress the colloidal crystal. Why do the authors suggest that the crystal strain is isotropic?

Our response:

Our conclusion is based on direct measurement of the particle positions. We find that in crystals grown on a flat substrate, the shape of the unit cell maintains a cubic shape, despite the variation of the unit cell size along the crystal's thickness. This is demonstrated in Fig. S1 (figure also shown above), where the profiles of the volume-per-particle $v(z)$ (Fig.S1a - blue) and the in-plane particle-particle distance $d_{||}(z)$ (Fig. S1b-blue) are plotted. The profiles satisfy $d_{||}(z) = (\sqrt{2}v(z))^{1/3}$, implying that under gravity, the unit cell deforms equally in the direction parallel and perpendicular to gravity. Importantly, this is no longer true for crystals grown on a template, as shown by the orange profiles in Fig. S1.

This result is not intuitive, as the reviewer points out. Suppose one considers an elastic body under normal stress or gravitational field. In this case the body is compressed in one direction, whereas it expands, rather than compresses, in the perpendicular direction, despite a decrease of the volume. From this perspective, colloidal crystals grown on a flat surface behave more like a fluid: gravitational field imposes changes in pressure, which translate into isotropic compression of the unit cell. This compression of unconstrained crystals in the transverse direction relative to gravity is accommodated by the grain boundaries and stacking faults, and the details of the mechanism will be discussed in a separate publication. The current manuscript briefly discusses isotropic compression in the Methods (Fig. S1 and section titled "Comparing sedimentation profiles of constrained and unconstrained crystals").

Reviewer #2:

2) The strain is associated with the crystal compression and compared to an unconstrained crystal. The type of dislocation, its energy, and its organization is presented to highlight the mechanism of crystal relaxation. The ionic strength is used to control the interparticle position. With the number of variables, It is difficult to identify which variables are key to the dislocation generation and plastic deformation.

Our response:

We thank the reviewer for pointing out this difficulty in the text. Our experiments provide detailed measurements of the plastic relaxation process in colloidal crystals. Our analysis maps the colloidal crystal to a classical thin film theory, which is based on linear elasticity. Therefore, once the mapping has been established, our conclusions on how dislocation interactions affect the relaxation process are general, and the details of colloidal suspensions do not play a role.

The reviewer correctly points out the key ingredients of this mapping:

(1) Definition of misfit strain. In the epitaxial growth of molecular crystals, the misfit strain is determined by the lattice spacings of the substrate and the film, $(a_f - a_s) / a_s$. In contrast, in colloidal crystals, the natural lattice spacing of the film is not unique and determined by the volume fraction of the particles. In the absence of gravity, the unit cell has a cubic shape. In the presence of gravity, we find that while the unit cell of crystals grown on a flat surface is indeed cubic, crystals grown on a template have tetragonal symmetry. As the two types of crystals have an identical volume fraction profile, we use particle-particle distance profiles measured on unconstrained crystals as a reference for strain in constrained crystals. Strain is defined with respect to a crystal with the same volume fraction but isotropic compression, in analogy to the definition of the deviatoric strain. Once this measure of strain is adopted, linear elasticity can be exploited as in molecular thin films.

(2) Control of misfit strain. In epitaxial growth of molecular crystals, the control of the misfit strain is usually achieved by variations of the molecular composition of the film. In colloidal crystals, the misfit strain depends on the crystal thickness and the contrast between the template spacing and the particle size. Fine control of the template and particle size is difficult to achieve. We therefore demonstrate an alternative approach to control lattice mismatch: By reducing the amount of salt, we increase the electrostatic repulsion between the particles and introduce slight softness of the inter-particle potential. We find that this softness can be mapped to a hard-sphere potential by considering a slightly larger particle size, as demonstrated in Fig. S2. This variation of the effective particle size allows us to vary the mismatch level systematically. We find no evidence that the softness of the potential plays any additional role.

(3) Variation of the strain along the thickness of the film. The variation of lattice spacing along the height of the crystal is unique to colloidal crystals. Our model accounts for the variation of strain and the resulting variation of the elastic moduli (Eq. 2), as described in detail in the Methods. Whereas this complexity does not allow us to obtain an analytical solution, it can be easily addressed numerically, as shown in the text. Importantly, ignoring this depth-dependence in the model does not change any of the qualitative behavior.

Once the misfit strain is defined, we demonstrated how dislocation interactions, such as the formation of Lomer-Cottrell and Hirth junctions, give rise to rapid relaxation and the formation of complex networks. The interaction mechanisms we have identified do not depend on the ionic strength or depth and are general. The physical control parameters determining the relaxation process are the misfit strain and crystal height.

Additional discussion regarding the generality of our results appears in our answers to the comments by reviewer 3.

We have rewritten the section “Modeling strain relaxation” in the Methods to clarify these points: To highlight the generality of our conclusions, we now introduce the classical thin film theory and contrast it with our model adapted to colloidal crystals. We have also modified the main text to address the reviewer’s suggestion and discuss the conditions under which our results can be generalized to atomic systems:

Line 268 - “We have demonstrated the role of dislocation interactions in the plastic relaxation of strained colloidal crystals. Our analysis shows that these interaction mechanisms can be readily transferred to atomic systems, despite some of the unique properties of colloidal crystals, such as the relatively simple hard-sphere interparticle interactions and height-dependent osmotic pressure (see Modeling strain relaxation section in \cite{Methods}). These interaction mechanisms are particularly relevant to systems that satisfy several conditions: First, plastic strain should be mediated by Shockley partial rather than perfect dislocations. Dissociation of perfect dislocations into partials \cite{Anderson_2017} in hard-sphere colloidal crystals \cite{Schall_2004,Schall_2006,Alsayed_2005,Lin2016,VanSaders_2018} is energetically favorable due to vanishing stacking fault energy \cite{Frenkel_1984}. Partial dislocations, however, are not unique to colloidal crystals and are common to low-stacking fault metallic and semiconductor films \cite{Maree1987}. Second, we have considered here tensile misfit strain, whereas we expect different relaxation mechanisms if the substrate imposes compression. Finally, our analysis builds on the equilibrium theoretical framework, suggesting that lattice resistance to dislocation nucleation and glide plays a secondary role and does not inhibit dislocation kinetics. Although in some atomic systems the lattice resistance is strong and can result in a metastable film growth well above the critical height, our results should apply when this kinetic barrier to dislocation motion is lowered by elevating the film's temperature.”

Reviewer #2:

3) A key result is the network Lomer-Cottrell dislocations which are a type of dislocation that form in crystalline materials with high stacking fault energy. Typically, they are a type of edge dislocation that are locked in place, but the authors show an evolution in the crystal and growth of dislocation as a function of crystal thickness. How do these dislocations break to accommodate the crystal strain?

Our response:

This is an interesting question. Lomer-Cottrell dislocations are indeed immobile as their Burgers vector is not on one of the (111) fcc easy glide planes. For this reason, for example, they play a crucial role in metal hardening by hindering the motion of mobile dislocations. Surprisingly, we find

that the formation of LC dislocations provides the dominant mechanism for mediating the relaxation process. The mechanism, however, does not require breaking the LC dislocations. Instead, strain is relaxed by elongation of LC segments: two Shockley dislocation threads glide and combine to form an LC segment, in a process similar to a zipping mechanism, as seen in Fig. 3C, 3E, and Mov. 3.

The nature of LC dislocations is discussed in the paragraph starting on line 147 and to clarify this process more, we modify the caption in Fig. 3c:

“A Lomer-Cottrell (LC) segment (red), $1/3[110]$, is formed by nucleation of a $1/6[11-2]$ loop in the vicinity of a pre-existing $1/6[112]$ misfit segment. Elongation of LC segments takes place by the glide of the Shockley threads.”

Reviewer #2:

4) What is the role of the areal size on crystal relaxation? The authors are normalizing the energy and strains over area, but it seems like the area should influence the dislocation generation. If the crystals are significantly larger (or smaller), will the ratio of the Lomer-Cottrell to Shockley dislocations change?

Our response:

The discussion about geometry is important, and we thank the reviewer for raising it. We should consider three primary length scales: crystal thickness, crystal lateral dimension, and the average length of dislocation segments. We grow crystals with a lateral dimension $\sim 10\text{mm}$, significantly larger than their thickness (up to $55\ \mu\text{m}$) and average segment length (up to $200\ \mu\text{m}$). Therefore, to an excellent approximation, the lateral boundaries of the crystals can be ignored, and in practice, the crystal is a thin film with an infinite lateral extent. However, we expect that the area will play a role if the lateral dimensions are similar to the crystal thickness. This limit, however, is difficult to achieve in our current experimental setup and has not been explored. As the measured total length of dislocation segments L depends on the imaging window ($200\ \mu\text{m} \times 200\ \mu\text{m}$) we normalize by that area to obtain dislocation densities (length per area). Similarly, in the model, the area can be scaled out by defining the dislocation density and energy density.

To emphasize the geometry, we modify the text:

Line 67 - “Thin colloidal crystals are grown over an area of 1cm^2 to a height of $h = 55\ \mu\text{m}$ at $5\ \mu\text{m}$ per hour, by sedimentation of the particles on either flat or templated substrates.”

Methods, Modeling strain relaxation:

Line 535 - “We consider strain relaxation in a thin film of an isotropic linear elastic material characterized by a shear modulus μ and a Poisson ratio ν . As the lateral dimensions of the film are much larger than its height and the length of dislocation segments, we ignore any lateral edge effects. The total elastic energy (per unit area) is approximated...”

Reviewer #3 (Remarks to the Author):

Review of “Dislocation interactions during plastic relaxation of epitaxial colloidal crystals”

In this study, the authors grow colloidal crystals (comprised of spheres) on templates and observe the growth and interaction of dislocations that relax the strain due to the template. The authors identify a two-step mechanism that accompanies relaxation; the nucleation of Shockley dislocations and subsequent merging into a Lomer-Cottrell dislocation. They provide convincing evidence that the growth of Shockley dislocations is the limiting step and requires large strain. Overall, I find the work to be of good quality and I have only minor comments (see below). The study appears to be carefully done and reasonably interpreted.

Our response:

We thank the reviewer for his/ her careful review of our work and are glad that the reviewer appreciates the quality of our work.

Reviewer #3:

However I am reluctant to recommend publication in the current state, as the study’s significance is under-developed. I can explain what I mean through the following points:

1. The main motivation given for this work is the importance of dislocations to the properties of thin-film materials with electrical and magnetic properties. The authors claim that dislocation interactions in epitaxial materials are poorly understood due to the difficulty of observing them directly, and propose that colloidal materials can be used as an easier-to-observe model system. I find this justification somewhat weak. Questions about dislocation evolution can be split roughly into two categories: (A) Long-ranged interactions where the material can be treated as a continuum or (B) Close-ranged interactions where the dislocation core structure is important. I do not believe we can learn very much about (B) in atomic systems from colloidal models, as the details of the interactions between atoms can be important. My interpretation is that this study falls into category (A). However I am left unsure what has been added to our understanding, aside from confirmation of preexisting simulation and theoretical results. I do not see how an experimental observation in a ‘model system’ is significantly different from a theory or numerical result - all three are models describing atomic systems. This is an issue of framing - the results obtained by the authors are interesting and significant for other researchers seeking to understand colloidal crystals, but are underwhelming as a demonstration of the ability of colloidal crystals to reveal new insights about atomic materials.

Our response:

We absolutely agree with the reviewer’s interpretation – our work falls into category (A): We aim to identify and understand general aspects of collective dislocation behavior, which do not depend on the exact interatomic interactions. The complex long-range mutual interactions between dislocations make crystalline plasticity a primary challenge in materials physics. At this moment, there is no unified theoretical framework that can bridge the gap between the dislocation dynamics and macroscopic material deformation; numerical simulations are often insufficient to fully capture the size of real systems; experiments do not allow detailed visualization of the

dislocation dynamics. Our work aims to fill this gap: we have a simple physical system that is fully controlled on which we can measure stress and strain and observe in-situ all microstructural changes down to the single particle level.

From a theoretical perspective, Cottrell [Dislocations in Solids Vol.11] argued that crystalline plasticity is “the most difficult remaining problem in classical physics” due to the complex dislocation interactions. Thin films open a window into the rather general aspects of collective dislocations dynamics, as the formed dislocation networks are relatively simple: in our case we start with a perfect single crystal and only four out of twelve slip systems are activated during the relaxation process. However, even with this apparent simplicity, accounting for the interactions between dislocations is challenging, and we still lack a theoretical description of the relaxation process. The attempts to propose different interaction mechanisms between dislocations often give contradictory results [Refs. 19-20], as some facilitate, and others hinder the relaxation. We would like to stress that our experiments do not provide confirmation of previous theoretical results, as suggested by the reviewer, simply because there is no established theory.

From an experimental perspective, the difficulty in atomic systems to simultaneously measure the evolution of the dislocation network and the corresponding macroscopic relaxation process hinders the identification of the relevant interaction processes. In this context, our experiments address this missing part and highlight new dislocation interaction mechanisms that have been ignored until now.

The reviewer rightfully contrasts our experiments with numerical simulations. Whereas numerical simulations can potentially supplement our experiments, at this point they cannot provide an alternative due to the high demand of computational power. For a simple estimate of the computational time needed to simulate our experiments, we can consider 5mmX5mm wide film, 50 μm thick, which roughly gives 500 million particles. Insisting on the lateral dimensions of 5 mm is important, as the sample should be considerably larger than both the film thickness and dislocation segments length (200 μm) to justify the thin film limit and avoid any boundary effects. Recent novel ultra-high-speed molecular dynamics (MD) simulations (Zepeda-Ruiz, Nature Materials 2021) have required extreme computational time (10^7 core hours) to simulate systems of ~ 300 million particles, even at extremely short time scales (~ 1 ns). Building on that, to simulate our system would require an unrealistically long period of time (~ 1 year) on a typical 1000 core cluster. An alternative modeling approach would be to simulate dislocations rather than atoms directly. The downside of these Discrete Dislocation Dynamics (DDD) simulations is that they require a significant phenomenological input to describe how dislocations nucleate, move, and multiply. For this reason, while it would be very insightful to compare DDD simulations with our experiments, these simulations do not provide a direct alternative.

It directly follows from the discussion above that our primary motivation is to identify universal aspects in the behavior of dislocations. The colloidal system allows us to address in full detail and in real-time the interplay between the mesoscopic and macroscopic scales - the dislocation dynamics and the relaxation of strain. To address the reviewer’s comment, we would like to clarify the findings of our work. Our work reveals several new dislocation interaction mechanisms and demonstrates their importance to epitaxial growth. First, we have identified the formation of Lomer-Cottrell dislocations and show that this process facilitates rapid relaxation of the misfit

strain. This result is surprising as the formation of LC junctions is usually associated with work hardening, during which these junctions hinder rather than facilitate slip. Second, we have identified different interaction mechanisms by which dislocation expansion can be blocked. We find that dislocations are blocked either by repulsion from perpendicular segments or by the formation of the famous Hirth-type junctions. These interactions give rise to complex structures within the dislocation network. The first blocking mechanism has been previously observed in atomic thin films and has been the basis for several theoretical models (Ref. 20). Therefore, an identical blocking mechanism in colloidal crystals demonstrates again the universal aspects of crystalline plasticity. In contrast, the formation of Hirth type dislocation threads has not been documented until now. Our ability to identify the formation of Hirth junctions explicitly demonstrates the crucial advantage of the colloidal systems: observing this interaction requires dynamic 3D visualization of the dislocation network, which is extremely difficult to achieve in atomic systems. These new insights can be directly mapped to atomic systems, as will be explained in our answer to the reviewer's second comment.

Finally, we would like to emphasize that although our motivation leans towards a more fundamental understating of collective dislocation behavior, this study is also important from a more practical perspective as dislocations play a crucial role in various technological applications. We will discuss some implications in answering the reviewer's third comment.

To better present our motivation we have modified the introduction of the paper:

Line 24 - "Atoms in crystalline materials are arranged in a perfect periodic order. Plastic deformation, which requires breaking this order, is mediated by nucleation and motion of topological line defects in the crystalline structure called dislocations \cite{Anderson_2017}. Due to the complexity of dislocation interactions, the collective behavior of these defects remains one of the principal challenges of materials and statistical physics \cite{Sethna_2017}. Of particular value, therefore, are fundamental experiments on simple systems in which all the elements (stress, strain and dislocation configurations) can be closely controlled and observed.

Here we focus on the mechanisms by which dislocations are formed in thin films. It has been observed that nucleation and growth of dislocations relaxes the elastic strain induced by the lattice mismatched substrate, if the crystals are grown above a critical thickness \cite{Matthews_1975,Nix1989,Kraft_2010}. The early stages of the relaxation process are well understood \cite{Matthews_1975,Houghton_1991,freund_1992}, as dislocations are well separated and their interactions can be ignored. During the later stages of relaxation, dislocation interactions play a crucial role; however, determining the interaction mechanism presents a significant challenge, with proposed mechanisms giving contradictory predictions \cite{Willis_1990,Willis_1991,freund_1992,Gillard_1994,Nix_1998,Stach_2000}. Our ability to identify the appropriate mechanism is limited by the difficulty to image in real-time both the relaxation of strain and the full 3D structure of the dislocation networks in atomic crystals.

Greater insight into thin film dislocation dynamics is also technologically important \cite{Tsaobook,Freundbook}: On the one hand, dislocations have detrimental effects such as decreasing electrical conductivity \cite{Chen2020}, photoconductivity \cite{Chen2020}, ferroelectricity \cite{Chu_2004}, thermoelectricity \cite{Zheng_2023}, and photonic band-gap \cite{Vlasov_2001}, while on the other hand, have been utilized to advantage to control the

functionality of thin films, enabling enhancement of superconductivity \cite{Dam1999,Llordes_2012}, switching of electrical resistance \cite{Szot_2006}, and fabrication of ordered nanostructures \cite{Brune_1998} and nanoscale ferromagnetic elements \cite{Sugiyama_2013}.”

Reviewer #3:

2. Most importantly, it is not clear to me reading this paper if the mechanism the authors describe (Shockley to Lomer-Cottrell misfit strain relaxation) is transferable to any atomic systems. What is the major cause of the trend of $\epsilon_e(h)$ shown in fig. 2A? Does the second term in eq. 2 contribute? At first glance it appears not to. Is this effect due then mostly to variations in $\epsilon_{tot}(z)$ and $\mu(z)$? This would seem to make the S to LC relaxation mechanism less general and maybe applicable only to colloid films under gravity. If this interpretation is incorrect I would urge the authors to discuss what atomic systems this result applies to.

Our response:

Our analysis implies that the relaxation mechanism we have identified is general and transferable to atomic systems. We thank the reviewer for bringing to our attention that this conclusion is not stated clearly enough.

To remedy this, we introduce first the classical misfit theory. This analysis doesn't account for the unique aspects of our colloidal system, such as the lattice constant variation with the crystal height and the image dislocations due to the stiff substrate. In this case, the total elastic energy (Eq. 2) can be schematically rewritten as:

$$\text{Eq. R1} \quad U \sim h \cdot \mu (\epsilon_{tot} - b_{||}L/A)^2 + L/A \cdot \frac{\mu b^2}{4\pi} \log(h/b)$$

The left term is the bulk elastic energy, which scales with the crystal thickness h . The elastic strain $\epsilon_e = \epsilon_{tot} - b_{||}L/A$ increases with the imposed ϵ_{tot} , and decreases with the growth of the dislocation density L/A . The right term is the energetic cost of introducing dislocations, and it scales with L/A and only weakly depends on h . Dislocation density is determined by competition between the two terms: the increase of L/A , on the one hand, decreases the first term and, on the other hand, increases the second term. Above a critical height h_c the first term becomes dominant, and energetically favorable to introduce dislocations. Therefore, to the reviewer's question, both the second and first terms play a crucial role in defining the critical height and the relaxation process.

Formally, L/A is determined by the minimization of the elastic energy $\partial U / \partial (L/A) = 0$, from which we can obtain the elastic strain

$$\text{Eq. R2} \quad \epsilon_e = \epsilon_{tot} - b_{||}L/A = \frac{1}{8\pi} \frac{b^2}{b_{||}} \frac{\log(h/b)}{h}$$

The critical height is determined by setting $L/A=0$. The increase of h above h_c results in a decrease in elastic strain. There is no theory that accounts for dislocation interactions and predicts the rapid relaxation process we have found. We use the classical theory separately for Shockley and Lomer-Cottrell dislocations and suggest that the rapid relaxation can be explained by the formation of Lomer-Cottrell junctions. It is evident from Eq. R2 that LC dislocations are more efficient in

mediating strain due to the lower value of the prefactor b^2/b_{\parallel} (and therefore lower residual ε_e): for Shockley $b^2/b_{\parallel} = a\sqrt{2}/2$ and for Lomer-Cottrell $b^2/b_{\parallel} = a\sqrt{2}/3$. Here b is the Burgers vector amplitude and b_{\parallel} is the Burgers vector in-plane component, where for Shockley $\vec{b} = a/6[1,1,\pm 2]$ and Lomer-Cottrell $\vec{b} = a/3[1,1,0]$.

Calculating the elastic energy is more intricate in colloidal crystals, as it requires taking into account the presence of image dislocations, and variations with z of the strains and elastic moduli, which precludes an easily interpreted form like Eq. R2. Nevertheless, despite this relative complexity, our numerical solution shows qualitatively similar conclusion – LC dislocations are more effective in mediating strain. The simplicity of Eq. R1 and R2 demonstrates the generality of this mechanism. Therefore, this relaxation mechanism, which we identify in colloidal crystals, should be readily transferable to atomic systems.

Which atomic systems should this interaction mechanism be relevant to? There are several underlying assumptions in the analysis presented above. First, the relaxation mechanism is mediated by partial rather than perfect dislocations. Partial dislocations are abundant in many atomic crystals. Therefore, our results can be directly mapped to crystals characterized by low stacking fault energy. The most famous examples are brass, silver, gold, and copper. However, we believe an analogous mechanism should occur in crystals with high stacking fault energies when two perfect, rather than partial dislocations, form Lomer rather than Lomer-Cottrell junctions. Second, our work focuses on thin films with a tensile rather than compressive misfit strain. We expect a different relaxation mechanism if the substrate imposes compressive strain, which we intend to address in the future. Finally, our analysis builds on equilibrium thin film theory. The excellent agreement between the measured and predicted critical heights suggests that our results should directly apply to systems that are not kinetically limited: the lattice resistance to dislocation nucleation and glide is relatively low. This is not the case in many semiconductors. The classical misfit theory often fails to predict the critical height as the crystals are often metastable and can be grown above h_c with no relaxation. The lattice resistance is dictated by the exact form of the interatomic potential. Nevertheless, it is well known that the classical theory is applicable to the same systems if the dislocation motion is facilitated by elevating the temperature. Hence, frustrated dislocation kinetics do not essentially limit the applicability of our results.

We have rewritten the section “Modeling strain relaxation” in the Methods to clarify these points: To highlight the generality of our conclusions, we now introduce the classical thin film theory and contrast it with our model adapted to colloidal crystals.

We have also modified the main text to address the reviewer’s suggestion and discuss the conditions under which our results can be generalized to atomic systems:

Line 268 - “We have demonstrated the role of dislocation interactions in the plastic relaxation of strained colloidal crystals. Our analysis shows that these interaction mechanisms can be readily transferred to atomic systems, despite some of the unique properties of colloidal crystals, such as the relatively simple hard-sphere interparticle interactions and height-dependent osmotic pressure (see Modeling strain relaxation section in \cite{Methods}). These interaction mechanisms are particularly relevant to systems that satisfy several conditions: First, plastic strain should be mediated by Shockley partial rather than perfect dislocations. Dissociation of perfect dislocations

into partials \cite{Anderson_2017} in hard-sphere colloidal crystals \cite{Schall_2004,Schall_2006,Alsayed_2005,Lin2016,VanSaders_2018} is energetically favorable due to vanishing stacking fault energy \cite{Frenkel_1984}. Partial dislocations, however, are not unique to colloidal crystals and are common to low-stacking fault metallic and semiconductor films \cite{Maree1987}. Second, we have considered here tensile misfit strain, whereas we expect different relaxation mechanisms if the substrate imposes compression. Finally, our analysis builds on the equilibrium theoretical framework, suggesting that lattice resistance to dislocation nucleation and glide plays a secondary role and does not inhibit dislocation kinetics. Although in some atomic systems the lattice resistance is strong and can result in a metastable film growth well above the critical height, our results should apply when this kinetic barrier to dislocation motion is lowered by elevating the film's temperature."

Reviewer #3:

3. In general, I find the consequences of this study under-explored. How can a researcher growing metallic or semiconductor thin films benefit from this work? With the current motivation of the paper, it seems like this is a logical and reasonable extension. On the colloidal side, where the effects are obviously applicable, how can they be exploited? While I am skeptical of colloids as atomic models, colloids do possess many interesting designable properties that aren't readily available with atoms. Examples include anisotropic shape, patchiness, or large responses to light/heat/solvent chemistry. Even something trivial, like density matching with the solvent - how might that impact the reported mechanisms? In this study, thin films are resistant to dislocation growth. A logical next question is, given that many properties of colloids can be tuned, how can the same resistance be extended to thicker (colloidal) films?

Our response:

We will address these comments separately.

3.1) In general, I find the consequences of this study under-explored. How can a researcher growing metallic or semiconductor thin films benefit from this work? With the current motivation of the paper, it seems like this is a logical and reasonable extension.

We would like to stress that we are motivated by the inherent difficulty in crystalline plasticity to bridge the gap between the macroscopic and mesoscopic scales. The deformation of thin films is an interesting mechanical problem as it allows us to explore the collective behavior of dislocations. The extended nature of these defects poses a significant theoretical challenge. However, despite our rather fundamental point of view, our work has important practical implications.

(1) We have demonstrated that the residual elastic strain is lower than expected from the classical thin film theory, which accounts for only one dislocation type. Determining the residual strain is a primary goal, as strain directly influences, for example, the electronic band gaps of semiconductor films. Therefore, accounting for a new mechanism that drives the relaxation process should help to improve the design of such systems.

(2) Our work has revealed an interplay between two types of dislocations: Shockley and Lomer-Cottrell. Dislocations are often used as easy paths for lateral transport, either of charges or of

dopants. Therefore, the presence of two types of dislocations and the difference in their core structure (see new Fig. S4) can be potentially exploited to fine-tune these transport processes.

(3) We have revealed how interactions between dislocations can block dislocation expansion and affect the structure of the dislocation network. In particular, we have demonstrated how the misfit strain and film thickness can be utilized to control the spacing and length of the dislocation segments independently. On the one hand, longer segments are beneficial when dislocations are used as a path for lateral transport. On the other hand, in fragmented networks, the high density of dislocation threads can be helpful in situations where dislocation threads are used as surface seeds for the growth of subsequent structures.

We thank the reviewer for motivating us to extend the discussion part of the manuscript:

Line 285 - “The comprehensive description of the relaxation process we provide here not only highlights the fundamental connection between dislocation interactions and the relaxation process, but also has practical implications. For example, accounting for the Shockley to Lomer-Cottrell transition that results in lower-than-expected residual strain should help improve the design of epitaxial semiconductor layers, as strain directly affects their band-gap. Furthermore, independent control of the spacing and length of the dislocation segments by misfit strain and film thickness can be utilized for defect engineering: longer segments are beneficial when dislocations are used as an easy path for the lateral transport of charges or dopants; the formation of fragmented networks with a high density of dislocation threads can be helpful surface seeds for the growth of novel micro- and nanostructures \cite{Brune_1998,Chen2020}. Finally, dislocations also play an important role in colloidal self-assembly \cite{Li_2021}. As assembly of colloidal crystals is a primary route for creating photonic band-gap materials \cite{Vlasov_2001}, the demonstrated control of dislocations could potentially be used to add different functionalities \cite{Hilhorst_2013}, such as creating optical propagation channels.”

3.2) On the colloidal side, where the effects are obviously applicable, how can they be exploited?

We thank reviewer #3 for motivating us to discuss applications in colloidal crystals. We follow the suggestions by reviewer #1 and have modified the introduction to include the negative effects of dislocations on the photonic band gap of colloidal crystals:

Line 41 - “Greater insight into thin film dislocation dynamics is also technologically important \cite{Tsao_book,Freund_book}: On the one hand, dislocations have detrimental effects such as decreasing electrical conductivity \cite{Chen2020}, photoconductivity \cite{Chen2020}, ferroelectricity \cite{Chu_2004}, thermoelectricity \cite{Zheng_2023}, and photonic band-gap \cite{Vlasov_2001}, while on the other hand, have been ...”

We now also discuss the implications of our results at the end of the manuscript, which includes the new opportunities in the context of photonic crystals that control of dislocations allows:

Line 295 - “Finally, dislocations also play an important role in colloidal self-assembly \cite{Li_2021}. As assembly of colloidal crystals is a primary route for creating photonic band-gap materials \cite{Vlasov_2001}, the demonstrated control of dislocations could potentially be used to add different functionalities \cite{Hilhorst_2013}, such as creating optical propagation channels.”

3.3) Colloids do possess many interesting designable properties that aren't readily available with atoms. Examples include anisotropic shape, patchiness, or large responses to light/heat/solvent chemistry. Even something trivial, like density matching with the solvent - how might that impact the reported mechanisms?

We agree with the reviewer that, in many ways, colloidal systems allow a more varied control of the system. For example, imposing strain in atomic thin films is challenging and often achieved by changing film composition. Instead, in colloidal crystals, we have gained excellent control by utilizing the density mismatch and varying the salt concentration. While density mismatch is essential to confine the crystal, decreasing the mismatch should not change the conclusions of our work: lower mismatch would reduce the misfit strain (if the same templates are used) and allow the growth of thicker dislocation-free crystals. Additional control parameters that can be implemented to impose strain and would be interesting to explore are swelling of polymeric particles or buoyancy changes due to changes in temperature.

The consequences of the anisotropic shape of the particles, suggested by the reviewer, would certainly be interesting to address. For example, we are eager to explore the growth of ellipsoidal particles. In this case, however, we expect the particles to form a nematic glass, which should have dramatically different behavior. Patchy particles, on the other hand, might form richer crystalline phases. New crystalline structures might invoke different types of dislocations, which in turn, would require a new analysis. Nevertheless, following the discussion presented in our answer to the previous two comments, we believe that our results are general and can be relevant to both more complex colloidal systems as well as various atomic crystals if several necessary conditions are satisfied – (1) The formed crystals have fcc structure and (2) are characterized by low stacking faults energy, which promotes the formation of partial dislocations. (3) Tensile rather than compressive misfit strain is imposed, and (4) dislocation glide is not limited by strong lattice resistance.

3.4) In this study, thin films are resistant to dislocation growth. A logical next question is, given that many properties of colloids can be tuned, how can the same resistance be extended to thicker (colloidal) films?

This is an interesting question. Although the relaxation process we observe cannot be explained by the current theoretical models, the measured critical strains are entirely consistent with the classical predictions. In fact, our experiments provide an unprecedented test of the theory in extremely thin films, up to 20 particle layers, where the applicability of the continuum theories should not be taken for granted. We, therefore, expect a decrease in the film strength with increasing thickness as long as the conditions described in our answer to the previous comment are satisfied - the same physics that make thin films extremely resistant makes thick crystals susceptible to dislocation growth. A possible way around that is to significantly increase the lattice resistance to dislocation motion by varying the interparticle interactions. In analogy to some semiconductors, this would frustrate the kinetics of dislocations and allow the growth of metastable thin films. These thoughts, however, do not build on our current results and are beyond the scope of this manuscript.

Reviewer #3:

Minor scientific points:

1. Line ~84: The authors observe that crystal films grown on flat substrates have isotropic compression of their units cells deep within the layer due to the weight of particles above. Isotropic compression is quite strange here - this is an anisotropic loading condition and one would expect anisotropic compression. The authors acknowledge that this is counter-intuitive and offer the explanation that it is “most likely due to the exchange of particles through the grain boundaries”. What does this mean? I don’t understand how that explains the observation.

Our response:

We are very grateful for the very careful reading of the text and Methods section. We indeed find this result intriguing and are currently working on a manuscript that explores in detail the microscopic mechanisms that give rise to isotropic compression.

In a perfectly ordered elastic crystal confined by rigid lateral boundaries, imposed normal stress would not cause any displacement of the particles in the perpendicular direction. Therefore, the in-plane lattice spacing remains unchanged, and the cubic unit cell becomes tetragonal. This behavior contrasts with a fluid, where the normal force translates to an isotropic compression.

Our measurements show that the in-plane lattice constant (perpendicular to gravity) of crystals grown on a flat substrate decreases with increasing normal load. In fact, we find (not presented) that the number of particles per layer is increasing. Squeezing in additional particles to each layer is impossible in a perfectly ordered crystal. The critical point, however, is that our crystals are imperfect and contain multiple grain boundaries and stacking faults, as shown in Fig. 1b. Our preliminary results show that the exchange of particles between the upper and lower layers takes place through either the grain boundaries or by formation or disappearance of stacking faults. For example, non-trivial dynamics of the staking faults can be observed in Movie S1.

We have slightly modified the Methods section to clarify this point, with the idea that these mechanisms are not essential for the completeness of the manuscript and will be discussed in detail in a separate publication:

Line 476 - “Compression in the direction perpendicular to the gravity axis is not intuitive and reflects the behavior of fluids and not single crystals. This compression requires squeezing particles from the upper to the lower layers of the crystal. While this exchange of particles is impossible in perfectly ordered crystals, it can be accomplished through the grain boundaries or formation and annihilation of stacking faults. Non-trivial dynamics of the staking faults during the sedimentation process are indeed observed in Movie S1. We will provide a detailed analysis of this process in a future publication.”

Reviewer #3:

2. Line 129: The authors note that thin films can be strained to larger values before plastically relaxing compared to thick films. As far as I can tell, these films are grown linearly in time at a rate of 5 layers per hour. This means that thick films are somewhat older than thin films. Does this

difference in total sample lifetime play a role here? Does the longer life of thick films allow for more time to nucleate dislocations?

Our response:

This is an interesting question that we have carefully examined.

First, we have verified that the difference in total sample lifetime does not play any role simply by doubling the sedimentation rate. Faster sedimentation is achieved by using sedimentation cells of half the height. Doubling the sedimentation rate enables us to compare experiments with a critical height of 40 μm to those with 20 μm . However, it is indeed interesting to reach even faster sedimentation in the future to explore how extreme sedimentation rates affect the growing crystals.

Furthermore, we have grown crystals to a height that is slightly below their critical height and found that the structure is stable with no evidence of dislocation nucleation. Interestingly, we can induce dislocations in these crystals by using high power of the confocal laser. The formed dislocations, however, quickly disappear once the laser power is reduced. These results demonstrate the robustness of the instability that takes place at critical heights.

We have modified the Methods section to clarify this point:

Line 439- "To allow slower sedimentation, typically of ~ 5 crystalline layers per hour, we double the cell height by sample extension made of polyethylene. We find no evidence for rate-dependence of the relaxation process if the crystal growth rate is doubled. Furthermore, no dislocation nucleation is observed if the final height of the grown crystals is kept slightly below the critical value. In fact, if dislocations are induced in crystals below the critical height, for example, by increasing the laser power of the confocal, they quickly disappear once the laser power is reduced."

Reviewer #3:

Small notes:

1. Line 85: It should be stated in the main text that $d_{\parallel}(h)$ is an average over all planes in the film, i.e. an average of $d_{\parallel}(z)$. I found the wording confusing in the main text.

Our response:

We thank the reviewer for pointing out this confusion. To clarify that, we modified the main text to:

Line 91 - "...Therefore, the decrease of v with h is reflected by the decrease of d_{\parallel}^0 , averaged over the thickness of the crystal, shown in Fig.1D (top) by the blue symbols."

Figure 1 caption now reads:

"(D, top) Evolution of d_{\parallel}^0 and d_{\parallel} , particle-particle distances in a plane parallel to the substrate, averaged over the crystal thickness, for crystals grown on a flat substrate and on a template, respectively. The onset of relaxation is marked by a sharp decrease of d_{\parallel} as h reaches a critical thickness $h_c \approx 26 \mu\text{m}$."

Reviewer #3:

2. Line 105: The authors state that “tensile strains result from contraction of the reference spacing $d_{||}^0$ with h , and do not involve any mechanical stretching of the sample”. This is confusingly worded - strain is a mechanical stretching away from a reference. I think it would be more clear to say that $d_{||}$ does not increase with h (if that is true).

Our response:

We have modified the text to clarify this point. Thank you.

Line 107 - “The total strains imposed by the templates in the [100] and [010] directions, which are equal, are given by $\epsilon_{tot} = (d_t - d_{||}^0)/d_{||}^0$ and increase with h , as shown by the blue symbols in Fig.1D (bottom). Importantly, during crystal growth, the imposed strains result from contraction of the reference spacing $d_{||}^0$ with h , and do not involve mechanical stretching of the template spacing d_t .”

Reviewer #3:

3. Fig 3: It is difficult to quickly understand the orientation of the camera in panels B-E. Can coordinate arrows (as in A) be drawn in the corner of each image?

Our response:

To clarify this point, we added an additional coordinate system in panel b. Thank you.

Reviewer #3:

4. Fig 5: The segment length l is difficult to understand from the label in panel B II. It is simply very hard to tell which of the dislocation crossings (for instance in panel B I) are endpoints. If none of the crossings are segment ends, then this should be stated. If some are, then maybe they can be marked on the plots somehow.

Our response:

This is a very important point, thank you. None of the crossings in panel B are endpoints of the segments. The apparent discontinuity of the segments when two dislocations cross is due to failure of the dislocation detection algorithm. This is easily confirmed by examining the 3D view and validating that at these positions there are no dislocation threads that go through the thickness of the sample, as shown in Fig. 3E. Importantly, we account for this failure of the algorithm in our analysis of the number of dislocation endpoints by counting the number of threads instead of the apparent segment endpoints in the 2D projection.

Fig. 3E caption: “The apparent discontinuity of the segments at the crossing points (bottom) is due to a failure of the dislocation detection algorithm.”

Fig. 5B caption: “Note that the apparent discontinuity of the segments at the crossing points in examples I and III is due to a failure of the dislocation detection algorithm.”

REVIEWERS' COMMENTS

Reviewer #1 (Remarks to the Author):

I would like to thank the authors for their careful consideration of the comments of all referees, and for adjusting and expanding the manuscript. I fully agree with the revision, which has improved this interesting manuscript. I am pleased to recommend the acceptance of the manuscript in Nature Communications.

Andrei V. Petukhov

Reviewer #2 (Remarks to the Author):

The authors have made substantial revisions to their manuscript. In particular, the addition of descriptions of the dislocation motion are much improved. The authors have addressed the previous comments adequately.

Reviewer #3 (Remarks to the Author):

Review of revised submission "Dislocation interactions during plastic relaxation of epitaxial colloidal crystals".

I feel that the changes made in response to the reviewer comments have greatly improved the match between the introduction/discussion and main results. The study now seems motivated in a way that is appropriate for the scientific work. I agree with the authors in their response letter - I also feel that this version is clearer and improved.

The major point I raised in my initial review was a perceived mismatch between the framing of the study and the scope of the results - the authors have done a fine job of addressing this. They have also tended to the minor points that I raised. I in particular appreciate the addition of a classical misfit theory section to the SI.

I recommend this work for publication. I have one minor suggestion for the discretion of the authors (I view it as optional w.r.t to the work's readiness to publish):

In the response letter, a nice discussion of the limits of particle simulations was included. However in the revised version, there is no reference to these limitations. I think a phrase about the "intense computational requirements of numerical metallurgy" and a reference to Zepeda-Ruiz, L.A., Stukowski, A., Opperstrup, T. et al. Atomistic insights into metal hardening. Nat. Mater. 20, 315–320 (2021) in the first paragraph would make sense.

Reviewer #1 (Remarks to the Author):

I would like to thank the authors for their careful consideration of the comments of all referees, and for adjusting and expanding the manuscript. I fully agree with the revision, which has improved this interesting manuscript. I am pleased to recommend the acceptance of the manuscript in Nature Communications.

Andrei V. Petukhov

Our response:

We thank the reviewer for helping us to improve the manuscript.

Reviewer #2 (Remarks to the Author):

The authors have made substantial revisions to their manuscript. In particular, the addition of descriptions of the dislocation motion are much improved. The authors have addressed the previous comments adequately.

Our response:

We thank the reviewer for helping us to improve the manuscript.

Reviewer #3 (Remarks to the Author):

Review of revised submission "Dislocation interactions during plastic relaxation of epitaxial colloidal crystals".

I feel that the changes made in response to the reviewer comments have greatly improved the match between the introduction/discussion and main results. The study now seems motivated in a way that is appropriate for the scientific work. I agree with the authors in their response letter - I also feel that this version is clearer and improved.

The major point I raised in my initial review was a perceived mismatch between the framing of the study and the scope of the results - the authors have done a fine job of addressing this. They have also tended to the minor points that I raised. I in particular appreciate the addition of a classical misfit theory section to the SI.

I recommend this work for publication. I have one minor suggestion for the discretion of the authors (I view it as optional w.r.t to the work's readiness to publish):

In the response letter, a nice discussion of the limits of particle simulations was included. However in the revised version, there is no reference to these limitations. I think a phrase about the "intense computational requirements of numerical metallurgy" and a reference to Zepeda-Ruiz, L.A., Stukowski, A., Opperstrup, T. et al. Atomistic insights into metal hardening. Nat. Mater. 20, 315–320 (2021) in the first paragraph would make sense.

Our response:

We thank the reviewer for helping us to improve the manuscript.

We have modified the first paragraph to address reviewer's suggestion:

“Due to the complexity of dislocation interactions, the collective behavior of these defects remains one of the principal challenges of materials and statistical physics \cite{Sethna_2017}. The wide range of time and length scales makes numerical modeling of dislocation dynamics computationally demanding \cite{Bertin_2020,Zepeda-Ruiz2021}. Of particular value, therefore, are fundamental experiments on simple systems in which all the elements (stress, strain and dislocation configurations) can be closely controlled and observed.”